# Spinal motor neuron pools may be partly driven by impulsive common inputs

Javier Yanguas Mayo[1], Alejandro Pascual Valdunciel[1] iD, Stuart N Baker[2] iD, Pablo Laguna[1,3] iD, Dario Farina[4] iD and Jaime Ibáñez Pereda[1,2] iD

[1]*Biomedical Signal Interpretation and Computational Simulation (BSICoS) Group, Aragon Institute of Engineering Research (I3A), IIS Aragon, Universidad de Zaragoza, Zaragoza, Spain*

[2]*Faculty of Medical Sciences, Newcastle University, Newcastle upon Tyne, UK*

[3]*Centro de Investigación Biomédica en Red CIBER-BBN, ISCIII, Madrid, Spain*

[4]*Bioengineering, Imperial College London, London, UK*

Handing Editors: Richard Carson & Mathew Piasecki

The peer review history is available in the Supporting information section of this article (https://doi.org/10.1113/JP290395#support-information-section).

**The Journal of Physiology**

**Abstract figure legend** A schematic overview of the proposed motor neuron drive framework. Unlike the traditionally assumed continuous common input (cCI), we propose that impulsive common inputs (iCI) constitute a key driver of motor neuron (MN) pool activity. This framework is validated through biophysically realistic computational simulations and experimental recordings from tibialis anterior (surface EMG) and flexor carpi radialis (intramuscular EMG) during isometric contractions. The results show that iCI can explain key features of the experimental recordings, such as the presence of sporadic events of synchronization. We also show that iCI drive the MN pool towards a non-linear behaviour,

This article was first published as a preprint. Yanguas Mayo J, Pascual Valdunciel A, Baker SN, Laguna P, Farina D, Ibáñez Pereda J. 2025. Spinal Motor Neuron Pools May be Partly Driven by Impulsive Common Inputs. bioRxiv. https://doi.org/10.1101/2025.08.13.670065

and its presence distorts the transmission of cCI. These results constitute a paradigm shift in the current understanding of motor control, by indicating the existence of different inputs than the traditionally assumed.

**Abstract** Spinal motor neurons serve as the link between the nervous system and muscles. As the final common pathway of the neuromuscular system, they receive inputs from both higher-level controllers and afferent pathways. It is often assumed that spinal motor neurons are primarily driven by continuous common inputs (cCI) within different frequency bands. Within this framework, the motor neuron pool behaves as a linear amplifier of the cCI. This implies that the frequency content of descending and spinal oscillatory signals is preserved and faithfully transmitted to the muscles; thus, the spectral content at the output of the motor neuron pool corresponds to that of the cCI. However, this framework overlooks the possibility that motor neurons could also be driven by impulsive common inputs (iCI), which can induce synchronization among them and disrupt the linear transmission of other synaptic inputs at the pool level. To test this hypothesis, computational simulations and experimental data from two different human muscles were used to characterize different aspects related to motor neuron spiking synchronization at the pool level. Our findings suggest that, indeed, iCI can account for relevant features observed in experimental data such as the presence of synchronization events at the pool level. We also observed that such impulsive inputs can affect the linearity in the transmission of cCI by the motor neuron pool. This study represents pioneering indirect evidence of the existence of iCI as inputs to motor neurons.

(Received 25 October 2025; accepted after revision 13 April 2026; first published online 23 April 2026)

**Corresponding author** J. I. Pereda: Biomedical Signal Interpretation and Computational Simulation (BSICoS) Group, Aragon Institute of Engineering Research (I3A), IIS Aragon, Universidad de Zaragoza, Zaragoza, Spain. Email: jibanez@unizar.es

**Key points**

- The current understanding of the motor control of voluntary movements assumes a continuous control, driven by oscillatory common signals.
- Some aspects of motor unit pool behaviour (particularly in terms of spiking synchronization and spectral content) typically observed in experimental recordings cannot be reproduced in simulations that only use continuous common inputs (cCI) to motor neurons.
- This study provides evidence indicating that spinal motor neurons receive a portion of their synaptic input in the form of impulsive common inputs (iCI) that synchronize their activity.
- The study also shows how such iCI can affect the linear transmission of other cCI by the motor neuron pool.
- These findings constitute a fundamental paradigm shift in the understanding of motor control and impact the development of interfaces that extract information from the activity of spinal motor neurons.

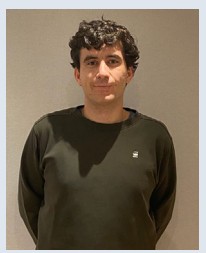

**Javier Yanguas Mayo** is a PhD candidate in the BSICoS research group at the University of Zaragoza. He obtained a bachelor's degree in Biomedical Engineering and a master's degree in Biomedical Research, with a specialization in Neuroscience and Cognition, from the University of Navarra. His doctoral research focuses on developing computational methodologies to extract and characterize neuronal sources from the action potentials generated by populations of motor neurons, combining biophysically realistic models with experimental data from healthy human subjects.

## Introduction

Spinal motor neurons (MNs) integrate inputs received from different parts of the nervous system and transmit this information to the muscles (Gandevia, 2001). MNs can be modelled as elements receiving two types of inputs: those shared across a pool of MNs (common inputs or CI) and those that are different for each MN (independent inputs or II) (De Luca & Erim, 1994; Farina et al., 2014). During constant-force muscle contractions, spinal MNs fire at relatively steady rates. In this context, the power spectral density (PSD) of the MN activity is largely dominated by the average discharge rate of the MNs and its harmonics. The power related to the inputs received by the MN (especially at frequencies above the discharge rate) are less prominent (Nakao et al., 1997). This observation is explained by the behaviour of individual MNs as non-linear systems. However, when the activity of groups of MNs receiving common inputs is summed, the CI are amplified proportionally to the total number of MN discharges. This characteristic of the MN pool to act as a linear amplification system of CI has been previously described and it applies to both low- and high-frequency inputs received by the MNs (Baldissera et al., 1998; Farina & Negro, 2015; Farina et al., 2014). The implication of this framework is that the summed activity of a sufficiently large number of MNs may be considered a reliable estimate of the net CI they receive. Because evidence exists of stable transmission of brain signals to the MNs, this also implies that the compound MN activity may be used to estimate CI from the brain (Conway et al., 1995; Halliday et al., 1998; Ibáñez et al., 2021).

The conceptual framework of MN pool acting as a linear amplifier of CI is valid only for continuous CI (cCI). This may partly be a result of the intuition that, for relatively steady forces, the neural inputs to MNs will also have slow continuous dynamics. However, there are many examples in the literature suggesting that a portion of the CI that MNs receive could be better modelled as a series of impulses or short-lived bursts. For example, intermittent submovements have been observed during slow-varying contractions (Karniel, 2013; Susilaradeya et al., 2019) and brief inhibitory-excitatory inputs to MNs have been detected when passively attending to external stimuli (Novembre et al., 2018). The frequently observed phenomenon of cortical beta bursts transmitted to the MN pool (Echeverria-Altuna et al., 2022) is a further example of intermittent inputs. These types of inputs may be sufficiently strong to align the firing times of different MNs transiently. This transient synchronization will impact on the linear behaviour of the MN pool: increased spike alignment across MNs will make the summed activity of the pool more similar to that of a single MN in terms of spectral distribution of power, with a dominance of the spectral peaks at the frequencies related to the discharge rate and multiples of it (i.e. harmonics).

The existence of impulsive CI (iCI) projecting to MN pools and their impact on MN behaviour has not been systematically addressed. Because these CI are brief and intermittent, their effects on MN activity are also restricted in time (short-lived events of synchronization). Importantly, standard methods used to analyze muscle signals are either based on the analysis of pairs of motor units or assume stationarity of the analyzed signals (Dideriksen et al., 2018; Laine & Valero-Cuevas, 2017; Negro & Farina, 2012; Ward et al., 2013). These methods are therefore not suitable to study non-stationary impulsive signals, which may explain why the existence of such iCI has not been analyzed so far. This gap has resulted in exclusive focus on cCI under stationary conditions, whereas the possibility that MNs receive non-stationary inputs, and how they respond to such inputs, remains uninvestigated. Characterizing MN responses to non-stationary inputs can have important implications for neural interfaces that decode motor intent from spinal motor neuron activity because it provides a more complete understanding of population level behaviour.

Here, we analyze experimental data from humans and compare it with computational simulations to verify whether MN pools receive iCI and also whether these inputs alter the linearity assumption in CI transmission by MN pools. The experimental results indicated that MN pool activity presents events of high synchronization at the population level, which agrees with simulation results testing the hypothesis that MN pools receive iCI. Additional tests also showed that the iCI alter the linear behaviour of the MN pool, affecting the linear transmission of cCI and therefore their estimation based on the measurement of motor unit activity.

## Methods

The primary aim of this study was to determine whether the activity of the MN pool is partly driven by impulsive-like input components. For this purpose, we combine the analysis of both experimental data from human subjects and data of three simulated scenarios that aimed to reproduce the experimental data. The subsequent sections describe the characteristics of the experimental data analyzed, the computational model and simulations used, and the procedures used to derive the results of the study.

### Human data

In this study, we analyzed experimental data from two different experiments recording isometric contractions

at 10% of the maximum voluntary contraction (MVC). The first experiment (from a previous study) involved recordings from the tibialis anterior (TA) muscle and the second experiment involved recordings from the flexor carpi radialis (FCR) muscle. Analyses were performed on two different muscles to avoid restricting the study to a single motor context, with muscles selected from anatomically distinct regions (upper and lower limbs).

Regarding the TA experiments, the characteristics of the dataset analyzed in this study have been described previously (Ibáñez et al., 2021). All subjects provided their written informed consent. The study was approved by the University College London Ethics Committee (Ethics Application 10037/001) and was carried out in accordance with the *Declaration of Helsinki*. The experiments involved EMG recordings from the TA muscle in 19 subjects (two females) during blocks of sustained isometric contractions at 10% of the MVC. Each subject performed two blocks and MN activity was decomposed from each block separately. Because tracking the activity of individual MNs across the two blocks was only possible in a reduced set of MNs, blocks were analyzed separately. For clarity, the main results presented here only consider, for each subject, the block from which the largest set of MNs was decomposed (results from the blocks with less MNs are comparable to the ones shown here and lead to identical conclusions).

The recorded EMG was obtained when subjects sat on a straight-backed chair with their knees flexed at 90° and their right foot positioned beneath a custom-made lever designed to measure ankle dorsiflexion forces during isometric contractions. Ankle force measurements, along with high-density electromyography (HD-EMG) recordings of the TA (using grids of 5 × 13 channels with an inter-electrode distance of 8 mm), were recorded using a multichannel EMG amplifier (Quattrocento, OT Bioelettronica, Turin, Italy). A sampling rate of 2048 Hz was used.

After a standardized warm-up, subjects performed three MVCs of the recorded muscle in each session, with at least 30 s of rest between trials. During MVCs, they were instructed to contract as forcefully as possible for at least 3 s. The highest recorded force across the three trials was taken as the final MVC value and digitally stored. Subsequently, subjects followed ramp-and-hold force trajectories displayed on a monitor. Each trial consisted of a 2 s ramp contraction from 0% to 10% MVC, followed by a 60 s holding phase at 10% MVC. During this phase, subjects maintained an isometric force at 10% MVC. Each subject repeated the ramp-and-hold contraction twice, with a 2 min interval between trials.

The second dataset was acquired from the FCR muscle during 10% MVC contractions using high-density intramuscular EMG (HDiEMG). Six subjects participated in the experiment after providing written informed consent. The study was approved by the local ethical committee

(CEICA, ID PI23-546) and conducted in accordance with the *Declaration of Helsinki*. A 16 channel, thin-film, intramuscular high-density electrode was inserted into the FCR muscle of each subject. The electrode array had a polyimide base with a linear configuration of platinum recording sites (each 5,257 $\mu m^2$) spaced 1 mm apart (Fraunhofer Institute, München, Germany). The electrode array was embedded in a 25 gauge needle used to insert the electrode in the muscle belly guided by ultrasound. Following accurate positioning, the needle was withdrawn, leaving the electrode securely in place (Muceli et al., 2015). Subjects were instructed to apply force towards a sensor flexing their right hand (FC22; Measurement Specialties, Hampton, CA, USA) during receipt of visual feedback of the applied force. The contraction followed a trapezoidal force profile consisting of a 2 s ramp-up phase, a 56 s steady plateau and a 2 s ramp-down phase. Force and HDiEMG signals were recorded simultaneously using a multichannel EMG amplifier (Open Ephys) (Siegle et al., 2017) at a sampling rate of 10 kHz. At the end of the experiment, the electrode array was removed from the muscle.

## Simulated data

In this section, we present the computational model used to simulate the activity of a MN pool and describe each simulation conducted with the model.

**Computational model overview.** The model used for the simulations is the one presented in Williams and Baker (2009). In short, the model simulates a MN pool consisting of 177 slow-type MNs. This represents the type of MN active during low-level muscle contractions. Each of the $j$-th MNs simulated, $j \in \{1, \ldots, 177\}$ is represented as a conductance-based two-compartment model (dendritic and somatic compartment), following Hodgkin–Huxley kinetics. It includes eight active conductances found in mammalian MN (somatic: $g_{Na}, g_{K\text{-}dr}, g_{Ca\text{-}N}, g_{K(Ca)}$ and $g_{NaP}$, dendritic: $g_{Ca\text{-}L}, g_{Ca\text{-}N}$ and $g_{K(Ca)}$). The membrane potential change in the somatic compartment is determined by:

$$C_m \frac{dV_S}{dt} = g_{Na}\, m_\infty^3\,(V_S)\, h\,(V_S - V_{Na}) + g_{K-dr} n^4\,(V_S - V_K)$$

$$+ g_{Ca-N} m_N^2 h_N\,(V_S - V_{Ca})$$

$$+ g_{K(Ca)} \frac{Ca_S}{Ca_S + K_d}\,(V_S - V_K)$$

$$+ g_L\,(V_S - V_L) + \frac{g_c}{p}\,(V_D - V_S) + g_{NaP} m_P^3 h_P$$

$$(V_S - V_{Na}) \tag{1}$$

The membrane potential change in the dendritic compartment is determined by:

$$C_m \frac{dV_D}{dt} = g_{Ca-N}\, m_N^2 h_N\, (V_D - V_{Ca})$$
$$+ g_{Ca-L} m_L\, (V_D - V_{Ca})$$
$$+ g_{K(Ca)} \frac{Ca_D}{Ca_D + K_d}\, (V_D - V_K)$$
$$+ g_L\, (V_D - V_L) + \frac{g_c}{1 - p}\, (V_S - V_D)$$
$$+ g_{excit}\, (V_D - V_{excit})$$
$$+ g_{inhib}\, (V_D - V_{inhib}) \tag{2}$$

where $V_s$ and $V_d$ are the voltage in the soma and dendrite compartments, and $p$ is the ratio of somatic surface area to total cell surface area, which was held constant in this study. The gating variable $m$ is the sodium-channel activation gate, the gating variable $h$ is the sodium channel inactivation gate and the gating variable $n$ is the potassium channel activation gate. Each gating variable is governed by an equation of the form:

$$\frac{dw}{dt} = \frac{w_\infty\,(V) - w}{\tau_w} \tag{3}$$

where $\tau_w$ is the activation and inactivation time constant, and $w_\infty(V)$ is the steady-state activation and inactivation function for the gating variable $w$, and it is given by:

$$w_\infty\,(V) = \frac{1}{1 + \exp\left[\frac{V - \theta_w}{k_w}\right]} \tag{4}$$

where $\theta_w$ is the half-voltage activation and inactivation function for the gating variable $w$. Similarly, $k_w$ is the activation and inactivation sensitivity for the gating variable $w$. The parameters used to solve the above equations were the same as those used in Williams and Baker (2009). The model included a population of 64 Renshaw cells (RC) that constitute the only inhibitory input to the MNs. Each MN received projections from 20 RC and each RC received projections from 50 MN. The membrane potential of the RCs also followed Hodgkin–Huxley kinetics. The RC had a mean firing rate of 11 spikes s$^{-1}$. The computational model also simulated force, as described in Williams and Baker (2009). In this model, the motor unit force was scaled by a gain factor, which depended on the unit's twitch contraction time and the interspike interval. Further details can be found in Williams and Baker (2009).

The sampling rate of the computational model was 5000 Hz. The differential equations governing MN membrane potential were solved using the exponential integration scheme (MacGregor, 2012), with a time step

**Table 1. Simulation scenarios (rows) as a function of the types of CI (columns) to the MN pool included in the simulation**

|  | Noisy II | Constant II | cCI | iCI |
|---|---|---|---|---|
| iCI scenario | X | X | — | X |
| cCI scenario | X | X | X | — |
| Control scenario | X | X | — | — |
| Combined CI scenario | X | X | X | X |

of 0.2 ms. The model was coded and run in MATLAB, version 2024a (The Mathworks Inc., Natick, MA, USA).

**Simulation scenarios.** In all the simulations, each of the 177 MNs was modelled as receiving a CI, $m\mathrm{CI}(t)$, composed of several components, and an independent input, $m\mathrm{II}(t)$, that was different for each MN. Therefore, the total input to each MN was defined as $m(t) = m\mathrm{II}(t) + m\mathrm{CI}(t)$.

The CI components depended on the simulated scenario, as explained below. The independent inputs $m_{II}(t)$ were simulated as a constant value $C_j$ plus a MN-specific individual runs of white Gaussian noise, with zero mean and a standard deviation equal to the MN-specific, $\sqrt{(C_j)}$ value, quarantining a spike variability within the mean interspike range. In this study, $C_j$ is a constant value for which the statistics across MNs determined the MN average discharge rate (DR). We fitted the model to capture the broad variability in MN firing observed in our experimental TA dataset. To do so, the firing rate of each MN was individually adjusted by varying the value of $C_j$, the constant current. In the simulated MN pool, the slowest MN exhibited a mean DR of 7.3 Hz, whereas the fastest reached a mean DR of 13.4 Hz.

The overall mean DR across the MN pool was 10.4 Hz, with a standard deviation of 1.4 Hz. This average aligns with previously reported values for low-level contractions of the TA muscle (Ibáñez et al., 2021). For each simulation scenario, we ran 10 simulations of 30 s each, unless otherwise stated. The results were averaged across the simulations. The different simulation scenarios used in this study are summarized in Table 1 and described below. Figure 1 provides a graphical representation of the simulated scenarios and the simulated inputs to the MN pool.

*iCI scenario.* The main hypothesis of this study is that the MN pool receives iCI components synchronizing the activity of individual MNs with each impulsive event. To analyze this hypothesis, we defined this CI component as a train of excitatory rectangular bursts (except during the initial and final 1 s intervals), $m_{CI}\,(t) = \sum_{l=1}^{L} \alpha\, p(t - lD + \tau_l)$

Here, $p(t)$ is a rectangular pulse (pulse function) of unit amplitude and duration of 5 ms. Each rectangular burst was scaled by a factor $\alpha$ large enough to depolarize, on average, 50% of the MNs, triggering an action potential within the burst duration and up to 3 ms after its end. The $1/D$ rate (fundamental frequency) at which the depolarizing bursts occurred was varied between 0 and 4 Hz in steps of 1 Hz across different simulations. To avoid that the bursts series were perfectly periodic, a random burst occurrence variability factor $\tau_l$ bounded between 0% and 20% of the corresponding mean interburst period was added. From a spectral point of view, this variability range implies that the harmonics of the burst frequency in the PSD have a smaller power than the fundamental burst frequency (Lyons, 2010). Each MN also received an $mII(t)$, and a constant value $C_j$, as defined previously.

In an additional unreported analysis, we simulated a similar scenario but using a train of excitatory Dirac delta pulses as the iCI (instead of a train of rectangular bursts). The results were similar to those presented in this study.

*cCI scenario.* As mentioned in the introduction, the classic approach when simulating the firing properties of MN pools has been to consider cCI without impulsive CI.

To assess this scenario, we ran simulations considering different inputs: (1) a sinusoid at a frequency matching the average DR of the simulated MNs (10.4 Hz, as in previous simulations, referred as *FDR*); (2) a sinusoid at 20 Hz (simulating beta oscillations) (Ibáñez et al., 2021); and (3) a sinusoid at 1 Hz.

To define these sinusoidal inputs, we generated independent realizations of white Gaussian noise processes with zero mean and standard deviation of 1. We then bandpass filtered these realizations with second-order Butterworth filters centred at the three frequencies simulated and with 1 Hz bandwidth. We normalized this filtered noise by dividing it by its root mean square value. We then multiplied this signal by the squared root of the target power to achieve, resulting in $mCI(t) = \alpha \cos(2\pi Fit)$, with $Fi \in \{FDR, 20, 1\}$ Hz. To determine the target power of this CI component, we used the maximum power that did not alter the coefficient of variation of the simulated force ($<0.5\%$ of change) for the case of a cCI at *FDR*. This target power was estimated in previous simulations. Each MN also received an $mII(t)$, including a MN-specific constant value $C_j$, as defined previously.

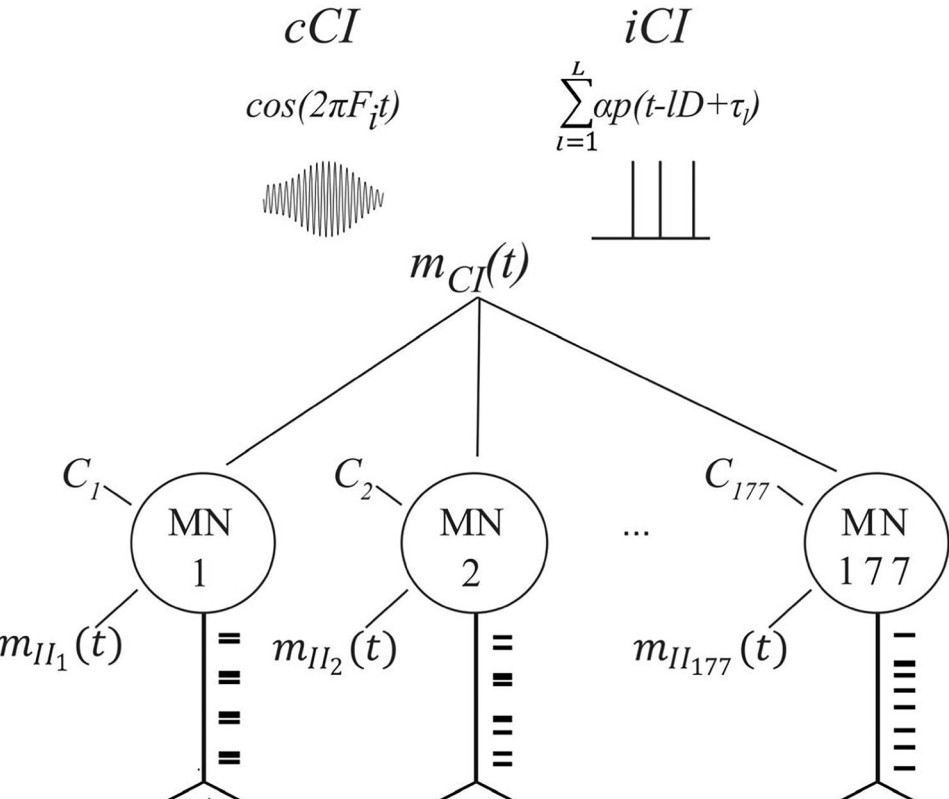

**Figure 1. Graphical representation of the computational model and the simulated scenarios**
$C_j$ represents a constant input, different for each $j$-th MN. $mCI(t)$ represents a common input for the MN pool that modulates the activity of the MNs. In our work, $mCI(t)$ was simulated as a sinusoid at a particular frequency (cCI) or as a series of impulsive pseudo-periodic bursts (iCI). $mII(t)$ represents the independent input, different for each MN.

In an additional analysis of this scenario (second section in the results), we used a sinusoid at *FDR* with a power scaled to match that of the harmonic term at *FDR* of the iCI at 1 Hz. In this complementary analysis, the iCI did not exhibit variability in the bursts occurrence times.

*Control scenario with no extra CI.* To explore the baseline behaviour of the MN pool in the absence of additional modulatory CI, we simulated a scenario in which the MNs did not receive any CI component (neither iCI nor cCI). This implies that the MNs only received $mII(t)$ including the MN-specific constant, $C_j$, that made them fire at a mean dominant rate, *FDR*, of 10.4 spikes $s^{-1}$ with the variance previously reported.

*Combined CI scenario.* Here, we simulated a scenario combining iCI and cCI. The objective was to analyze how impulsive inputs affect the ability of the MN pool to transmit cCI described in the current framework. We ran a set of simulations in which the MN pool received a CI formed by two different components: a sinusoid at either 20 Hz or 1 Hz, an iCI with a fundamental frequency of $1/D = 1$ Hz. We used a cCI at 20 Hz because there exists evidence of the transmission of such kind of activity across the spinal MNs (Bräcklein et al., 2022; Zicher et al., 2023). We used a cCI at 1 Hz to study the effects of the iCI in the frequencies related to the force control. The resulting common input becomes:

$$m_{\mathrm{CI}}(t) = \alpha_1 \cos(2\pi \mathcal{F}_i t) + \sum_{l=1}^{L} \alpha_2 p(t - l1 + \tau_l) \quad (5)$$

where *F*i is either 20 Hz, simulating the beta oscillations, or 1 Hz, simulating the force control signal. We tested different power levels corresponding to $\alpha_1$ to examine how the non-linear effects triggered by the iCI depend on the strength of the cCI projection to the MN pool. Specifically, we varied the power in 10% increments relative to the maximum power used. This maximum was defined as the highest power level that did not alter the coefficient of variation of the simulated force by more than 0.5%, when using a cCI of 20 Hz. The iCI was defined as in the previous section. The power of each pulse, controlled by $\alpha_2$, was sufficient to depolarize a mean proportion of 50% of the MNs, as simulated previously. In our tests, we compared results with and without including the iCI. In this context, we studied how the impulsive events affect the transmission of continuous signals at the known beta band and whether these effects depend on the power of the cCI component.

## Signal analysis

The signal analysis includes different blocks from pre-processing to the estimation of different metrics. Each block is described below.

**Preprocessing of experimental datasets.** For the TA dataset, for each subject, a data segment of 30 s corresponding to a period with a continuous constant 10% MVC force was extracted (this ensured that the decomposed MNs within that 30 s segment were firing steadily). The details about EMG decomposition are reported in the original study (Ibáñez et al., 2021).

Briefly, in the offline analysis, the HD-EMG signals were first bandpass filtered (20–500 Hz) using a second-order zero-lag Butterworth filter. The signals were then decomposed into motor unit spike trains $MN_j$, using blind source separation techniques (Holobar et al., 2014). Manual editing of the decomposed activity was conducted in accordance with previously described procedures (Hug et al., 2021). The *i*-th spike time $t_{i,j}$ of each *j*-th MN was defined as the onset of the motor unit action potential. To estimate this onset, double differential signals were computed from the monopolar HD-EMG recordings along the columns of the electrode grid, approximately aligned with the longitudinal direction of the muscle fibers.

For the FCR dataset, the iEMG signals were first bandpass filtered (100–4500 Hz) and then decomposed with a previously validated algorithm (Grison et al., 2024) and manually edited with DEMUSE, version 6.3 (The University of Maribor, Maribor, Slovenia) following the same edition procedure as employed for the TA data. MNs showing pulse-to-noise ratio above 30 dB were kept for the subsequent analysis. In line with the previous dataset, we selected for each subject the data segment of 30 s maximizing the number of motor units firing steadily.

For each MN decomposed, we computed its instantaneous firing rate in non-overlapping 1 s windows. MNs with a coefficient of variation in the instantaneous firing rate larger than 0.3 (Pascual-Valdunciel et al., 2025) were excluded from subsequent analysis. This exclusion criterion was implemented to exclude MNs that were not continuously recruited throughout the considered 30 s window of analysis. In this way, MNs for which spike trains intermittently appeared and disappeared (indicative of repeated recruitment and derecruitment near threshold) were excluded, whereas MNs that discharged consistently during the isometric contraction were retained. Given that the SPIKE distance metric depends critically on interspike interval patterns within decomposed spike trains, excluding MNs being recruited and derecruited repeatedly was essential to prevent artifactual distortion and noise in the calculated SPIKE distance values.

**PSD.** For both simulated and experimental data, the PSD of the cumulative spike train (i.e. the summed activity of all MNs) was computed using Welch's method (MATLAB function *pwelch*) using 1 s windows with an overlap of 0.5 s and a frequency resolution of 0.25 Hz. The

cumulative spike train was detrended before computing the PSD to remove the spectral component at 0 Hz. The simulated data included larger number of MNs than those obtained when decomposing experimental data. To compare experimental and simulated results, the cumulative spike train obtained from the simulated data was computed using 30 randomly selected MNs from the simulated data and this procedure was repeated 80 times, each with a different set of 30 simulated MNs, and the resulting PSDs were averaged. When computing the cumulative spike train in each iteration, we also computed the input-output gain at *FDR* averaged across iterations. Gain was defined as the ratio between the power at *FDR* of the cumulative spike train and the power at the same frequency in the common input $mCI(t)$.

**Intramuscular coherence (IMC).** The IMC is a widely used metric to estimate shared inputs across MNs within a pool (Dideriksen et al., 2018). Because IMC is calculated by averaging the spectral activity of subpools of MNs across different time segments, it serves as a useful tool for estimating stationary inputs to the MN pool. In other words, the IMC is sensitive to inputs that influence MN activity homogeneously over time, such as cCI.

Briefly, to obtain the IMC, spectral coherence is computed between the activities of two equally sized MN subpools and the results are averaged across multiple iterations with different subpools of MNs. In this study, magnitude-squared coherence was computed using Welch's method (mscohere MATLAB function), with 1 s windows, 0.5 s overlap and resolution of 0.25 Hz. For the experimental data, we first defined two subpools of MNs with half of the total number of MNs available for each subject (i.e. the maximal number of MNs per subpool to compute the IMC). We built the cumulative spike train of each subpool, detrended them and computed the coherence. This process was performed in 80 iterations and the coherences for each iteration were averaged. In each iteration, the MNs belonging to each subpool were randomly selected from the total set of MNs. For the simulated data, 15 MNs in each subpool were randomly selected from the full simulated MN pool (177 MNs) in each iteration (with the two permuted subpools, this implies a total pool of 30 MNs at each iteration). The same procedure was then applied to compute coherence across 80 iterations and the results were averaged.

**SPIKE distance.** The SPIKE distance (Kreuz et al., 2013) is a metric to quantify synchrony between spike trains for both simulated and real data. The SPIKE distance is a time-resolved measure of dissimilarity between spike trains that focuses on the precise timing of spikes rather than overall spike counts or firing rates. At each moment in time, it assesses how close the nearest spikes are across spike trains by considering both the preceding and following spikes. These instantaneous differences are then normalized by the local interspike intervals, allowing the metric to become independent of variations in firing rates and providing a sensitive, temporally detailed comparison of spike timing. Details can be found in Kreuz et al (2013), although the formula for the time-varying SPIKE distance between two spike trains (here, $c_n(t)$; $n \in \{1, 2\}$) is computed by defining the timing, $t_P^{(n)}(t)$ of the preceding spike to time instant $t$ in the spike train $n$ as:

$$t_P^{(n)}(t) = \max_{i, j_n} \left( t_{i, j_n} t_{i, j_n} \leq t \right) , \; j_n \text{ indexing MU in } c_n(t) \quad (6)$$

and the time instant following time t as:

$$t_F^{(n)}(t) = \min_{i, j_n} \left( t_{i, j_n} t_{i, j_n} > t \right) , \; j_n \text{ indexing MU in } c_n(t) \quad (7)$$

from where an interspike interval from later preceding to earliest following spikes at time $t$ can be defined as

$$T_{PF}^{(n)}(t) = t_F^{(n)}(t) - t_P^{(n)}(t) \quad (8)$$

and the difference between spikes timings at different sub pools for the preceding and following ones can be defined as:

$$\Delta t_P(t) = \left| t_P^{(1)}(t) - t_P^{(2)}(t) \right| \text{ and } \Delta t_F(t)$$

$$= \left| t_F^{(1)}(t) - t_F^{(2)}(t) \right| , \quad (9)$$

respectively. This definitions lead to propose a dissimilarity spike trains as the average of the inter-pool spike differences normalized by the mean interspike distance:

$$\frac{\Delta t_P(t) + \Delta t_F(t)}{T_{PF}^{(1)}(t) + T_{PF}^{(2)}(t)} \quad (10)$$

Aiming to be more local in time, giving more weight to interpool spike differences closer to analyzing time $t$, defining:

$$\delta_P^{(n)}(t) = t - t_P^{(n)}(t) \text{ and } \delta_F^{(n)}(t) = t_F^{(n)}(t) - t, \quad (11)$$

and weighting inversely to the across subpools mean of these distances, the following dissimilarity measure is obtained (Kreuz et al., 2013).

$$S(t) =$$

$$\frac{\Delta t_P(t) \left[ \delta_F^{(1)}(t) + \delta_F^{(2)}(t) \right] + \Delta t_F(t) \left[ \delta_P^{(1)}(t) + \delta_P^{(2)}(t) \right]}{\left( T_{PF}^{(1)}(t) \right)^2 + \left( T_{PF}^{(2)}(t) \right)^2} \quad (12)$$

The SPIKE distance $S(t)$ is a time-varying metric that can be computed in a multivariate way, by averaging the spike distances of each pair of spike trains. The SPIKE distance is a dissimilarity index: when the MNs are more synchronized, the index shows smaller values, bounded

to the interval [0, 1]. To implement the SPIKE distance in MATLAB, we used the code developed in the study by Kreuz et al (2013).

We aimed to relate the SPIKE distance to the population activity of the MN pool. As stated in the Introduction, individual MNs are governed by their DR. A synchronized pool of MN is also partially dominated by the DR of the pool, as the MNs are firing simultaneously. This implies that spectral power at $F$DR serves as an indicator of the pool's synchrony. Therefore, we calculated the SPIKE distance at time points corresponding to increases in the DR-band power of the cumulative spike train. This measure of synchronization based on the power at the $F$DR was applied to all three simulated scenarios and the experimental data, allowing for a comparison to identify which scenario most closely resembles the recorded activity.

To analyze this, the cumulative spike train was band-pass filtered (second-order Butterworth, 4 Hz bandwidth centred at $F$DR) and the instantaneous amplitude was extracted using the Hilbert transform. Time points where the amplitude exceeded the mean plus 1 SD were identified as synchronization events. Centred around each $k$-th event timing $t_k$, the time of maximal instantaneous amplitude at the filtered cumulative spike train, a 2 s window was extracted, and the SPIKE distance was averaged within that $k$-th window as an estimate of MN synchronization during periods of elevated power of the cumulative spike train at $F$DR.

As before, for the simulated data, 30 MNs were randomly selected in each iteration, and the SPIKE distance was computed for that subpool. This process was repeated over 80 iterations. In each iteration, the SPIKE distance was averaged across all the $k$-th windows centred on synchronization events and across the random time windows. These values were then averaged across iterations. For the experimental data, the same procedure was applied using all decomposed MNs for each subject.

To express the magnitude of the synchronization event, we computed the minimal value of the averaged SPIKE distance. We picked the minimal SPIKE distance value because the SPIKE distance is a normalized measure (bounded to the interval [0, 1]) and thus can be compared across conditions. To assess whether the experimentally observed SPIKE distance truly resulted from the influence of the described iCI rather than from the intrinsic statistical properties of the data, we conducted an additional analysis using surrogate data. For each subject, we generated a surrogate dataset by circularly shifting each spike train by a random duration of between 2 s and 8 s, drawn from a uniform distribution. SPIKE distance was then computed following the same procedure as with the original data (averaging values within the windows centred around peaks in the filtered cumulative spike train). This process was repeated over 100 surrogate datasets per subject, with the resulting SPIKE distances averaged across iterations. If the reduced SPIKE distance (increased synchrony) in the experimental data were solely a result of the intrinsic statistical properties of the dataset, then the minimum SPIKE distance observed in the surrogate data should closely match the experimental value. Conversely, if the reduction reflects the influence of an underlying input, a clear difference between the minimum surrogate and experimental values should be expected.

**Pearson correlation coefficient.** The correlation coefficient was used to quantify the cCI transmission across the MN pool in the final part of the study (where simulations combining impulsive and sinusoidal inputs were used). To account for potential phase shifts or lags between the signals, we computed the cross-correlation function between the CI and the filtered cumulative spike train and select the maximal value of this function at each iteration. In each simulated scenario, the correlation was computed using cumulative spike trains composed of either 10, 30 or 177 MNs, to assess transmission when considering smaller subpopulations *vs.* the entire MN pool. For the cases of 10 and 30 MNs, we performed a total of 80 iterations. At each iteration, the MNs were randomly picked from the entire pool and the spike trains of the selected MNs were summed. The resulting cumulative spike train was bandpass filtered in the same frequency band as the frequency used to generate the cCI using a second-order Butterworth filter of 1 Hz band-width centred at the frequency of interest. The correlation coefficients (maximal value of the cross-correlation function) were averaged across iterations.

## Statistical analysis

The central premise of this paper is that MNs do not receive information solely in the form of cCI. To determine whether the experimentally observed magnitude of the synchronization events is significantly greater than that observed under the influence of cCI at different frequencies, or in the absence of any inputs (reflecting the intrinsic behaviour of the simulated pool), a one-way ANOVA was performed between the minimal values in SPIKE distance for each subject in the FCR and in the TA muscles, and that of each simulation condition (one factor with seven levels). Because the assumption of homoscedasticity was violated (assessed by Levene test), a classical one-way ANOVA was inappropriate. Therefore, we used Welch's one-way ANOVA, which is robust to heterogeneity of variances across groups. *Post hoc* pairwise comparisons were subsequently conducted using the Games–Howell procedure, which likewise does not assume equal variances. This analysis was performed in

R, version 4.4.0 (R Foundation, Vienna, Austria) because MATLAB does not incorporate this functionality. When comparing the experimentally observed SPIKE-distance values in FCR and TA with the surrogate values, we applied a *t* test for the FCR data and a Wilcoxon rank-sum test for the TA dataset because the latter did not satisfy normality according to the Lilliefors test. For statistically significant comparisons, effect sizes (denoted by |*d*|) were calculated using Cohen's *d*. All of the results are reported as the mean ± SD.

## Results

This section is divided in three parts. First, we compare the experimental data with simulated scenarios to look for evidence supporting the hypothesis that MN pools receive iCI. Then, in the second and third parts, we show how iCI affect the transmission of cCI by MNs and how this may limit the estimation of these inputs from MN outputs. For the comparison of the metrics with the experimental data and for the transmission analysis (the first and third sections of the results), we used the case of iCI having a dominant rate $1/D = 1$ Hz (1 burst s$^{-1}$) because this was approximately the mean rate observed in the experimental data.

Regarding the experimental data from the TA dataset, the average number of MNs decomposed per subject included in the analysis was $22.8 \pm 7.3$ (range $10-35$). The average firing rate *F*DR of the MNs per subject was $10.5 \pm 1.2$ spikes s$^{-1}$.

Regarding the FCR dataset, the average number of MNs per subject was $15.2 \pm 4$ (range $11-20$). The average firing rate *F*DR of the MNs per subject was $12.9 \pm 2.3$ spikes s$^{-1}$.

### Simulations considering iCI reproduce key characteristics of MN pool activity recorded experimentally

An example of the simulated MN activity in the scenario using depolarizing iCI is shown in Fig. 2*A* (top). It is shown how MNs synchronize their firings every time an impulse occurs. As a consequence of this synchronization, the amplitude of the summed MN activity filtered at *F*DR increases briefly (Fig. 2*A*, bottom). This increase in power at *F*DR reflects that the MN pool is momentarily behaving more similarly to a single MN (where the power spectrum is dominated by a peak at the fundamental frequency, i.e. *F*DR). Similar dynamics may be qualitatively observed in experimental data (Fig. 2*B*, data from the TA of a representative subject), with sporadic alignments of MN firings associated with increases in the power of the cumulative spike train in frequencies around *F*DR.

Based on the above reasoning and aiming to identify moments of synchronization (reflecting iCI), we looked

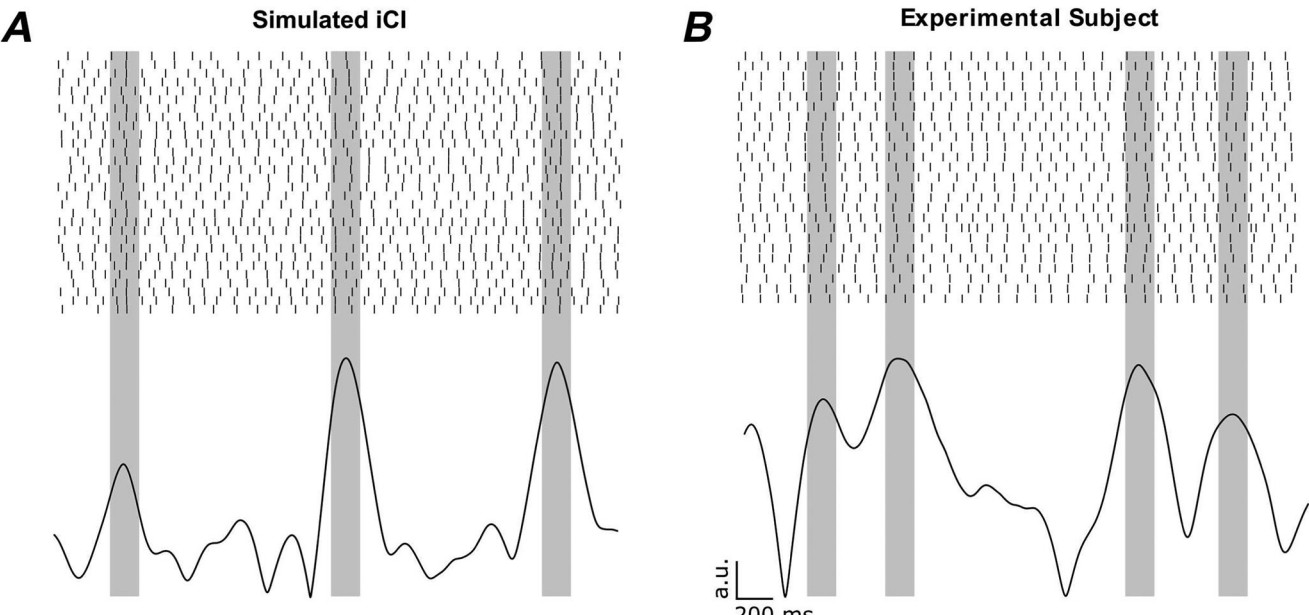

**Figure 2. Both, experimental recordings and iCI simulations, exhibit synchronized MN activity, indicated by increments in the instantaneous amplitude of the filtered CST**
*A*, raster plot (3 s) and instantaneous amplitude (obtained with the Hilbert transform) of the cumulative spike train of a simulated population of 30 MNs receiving an iCI of 1 burst s$^{-1}$, synchronizing its firings at the times of the bursts. *B*, raster plot (3 s) and instantaneous amplitude (obtained with the Hilbert transform) of the cumulative spike train of a population of 25 MNs from the TA muscle of a representative subject. The plots show how the peaks of the filtered cumulative spike train indicate an alignment of the MN firings.

for the time instants at which the power at the average *F*DR of the summed MN activity was high (more than 1 SD above the mean) and located the peak times of these events. For the experimental TA data, the average rate of the detected events with high power at the *F*DR frequency was $0.74 \pm 0.13$ events s$^{-1}$. When using the surrogate datasets, we did not observe a substantial difference in the number of detected events ($0.69 \pm 0.04$) with respect to the true dataset. However, the magnitude of the surrogate SPIKE distance was much less pronounced relative to that of the true dataset (discussed below). A similar pattern is observed with the FCR data, where the true rate of events is $0.66 \pm 0.23$ and the rate of the surrogate data is $0.70 \pm 0.08$, with a substantially greater magnitude in the true SPIKE distance relative to the surrogate. In the case of simulations using iCI, the mean rate was $0.87 \pm 0.11$ events s$^{-1}$. In the cases where simulations did not include cCI or they did include cCI, the rates were $0.64 \pm 0.11$ events/s for the cCI at *F*DR; $0.78 \pm 0.04$ events s$^{-1}$ for the cCI at 20 Hz; $0.73 \pm 0.13$ events s$^{-1}$ for the cCI at 1 Hz; and $0.75 \pm 0.09$ events s$^{-1}$ for the no-CI scenario.

Figure 3 shows the MN synchronization profiles (measured with the SPIKE distance metric) around the events of high power at *F*DR. Figure 3*A* shows the results obtained from the TA muscle, both for the true recorded data (averaged across subjects, blue curve, with grey curves representing individual subjects) and for the surrogate data (averaged across subjects, yellow curve). For the true experimental data (blue curve), the synchronization profile presented a short-lasting marked reduction (negative peak) around 0 s, reflecting that the moments of high power at *F*DR are associated with a sharp and brief increase in spike synchronization across MNs, as predicted. When performing the surrogate approach (by circular shifting each spike train by a factor between 2 s and 8 s and averaging across 100 iterations per subject), the average minimal SPIKE distance obtained was 0.285, which is significantly higher than the average minimal SPIKE distance of the true data, 0.254 ($P < 0.001$, $|d| = 2.66$). These results indicate that the synchronization observed in the TA data (blue curve) cannot be attributed to the intrinsic statistical structure of the dataset. Analysis of the FCR data (Fig. 3*B*) revealed a comparable pattern. The mean minimum SPIKE distance for the true experimental recordings (blue curve) was 0.243, significantly lower than the minimal value of the surrogated data (0.275, $P = 0.004$, $|d| = 2.55$). These findings indicate that the level of MN synchronization is significant and consistent across the examined muscles (FCR and TA).

The omnibus Welch's ANOVA between experimental and simulated conditions revealed a statistically significant difference ($P < 0.001$) in the minimum SPIKE

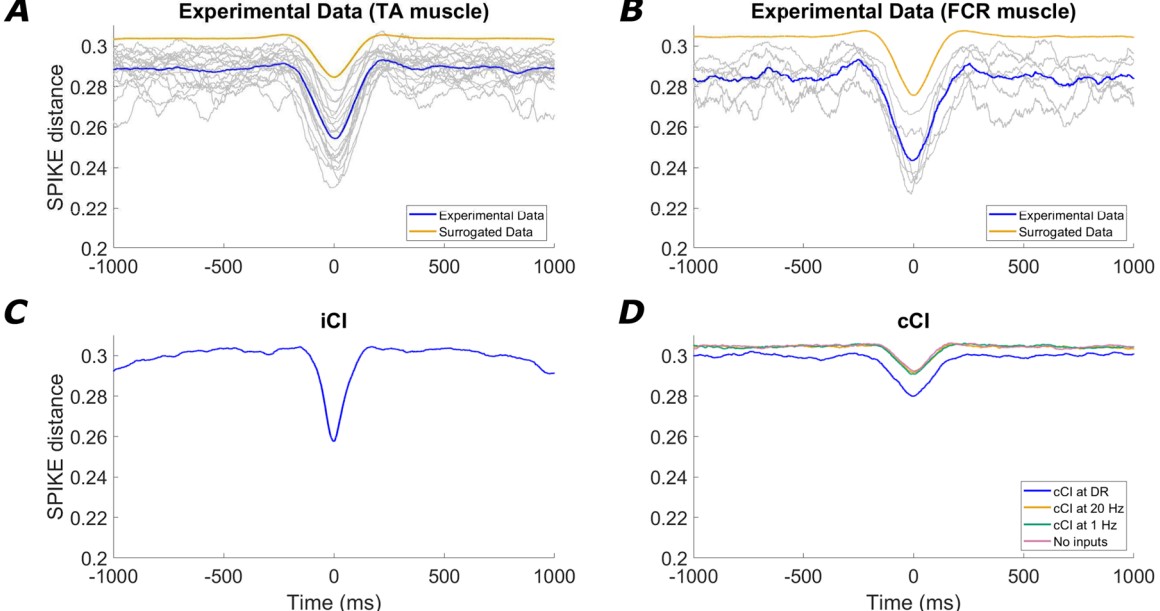

**Figure 3. Average SPIKE distance changes in segments around time points where the power of the summed MN activity presents a maximum at *F*DR**
The results obtained using experimental data (*A* and *B*, grey curves representing individual subjects and colored curves representing the mean of the true data, blue, and of the surrogate data, yellow), and simulations with iCI (*C*), cCI (*D*) and no-CI (*D*). A clear resemblance in MN synchronization is shown between the experimental data and simulations using iCI. The other scenarios do not lead to comparable results because they do not show a clear evidence of synchronization changes linked with increases in power of the summed MN activity at *F*DR.

**Table 2. *P* values and effect sizes from *post hoc* comparisons of minimal SPIKE distances, with rows indicating experimental data and iCI simulations and columns indicating cCI simulations and no inputs simulations**

|  | TA | FCR | iCI |
|---|---|---|---|
| cCI - 1 Hz | $P < 0.001$ | $4.00 \times 10^{-3}$ | $P < 0.001$ |
|  | $|d| = 3.04$ | $|d| = 5.15$ | $|d| = 6.82$ |
| cCI - *FDR* | $P < 0.001$ | $1.10 \times 10^{-2}$ | $P < 0.001$ |
|  | $|d| = 2.08$ | $|d| = 3.83$ | $|d| = 4.10$ |
| cCI - 20 Hz | $P < 0.001$ | $3.00 \times 10^{-3}$ | $P < 0.001$ |
|  | $|d| = 3.09$ | $|d| = 5.16$ | $|d| = 6.75$ |
| No CI | $P < 0.001$ | $3.00 \times 10^{-3}$ | $P < 0.001$ |
|  | $|d| = 3.16$ | $|d| = 5.28$ | $|d| = 7.05$ |

Note that the smaller (but significant) *P* values for the FCR condition are influenced by the reduced sample size in this condition ($n = 6$ subjects).

distance across all conditions, and *post hoc* analyses were conducted to compare the experimental and simulated data. Figure 3*C* represents the SPIKE distance from simulated data using iCI (rate of 1 burst s$^{-1}$), averaged across simulations.

Interestingly, this scenario also reproduces the short-lasting marked reduction (negative peak) around 0 s and overall shows a very similar pattern than that observed in the experimental data (blue curves Fig. 3*A* and *B*). In particular, the SPIKE distance for the simulated iCI showed an average minimal value of 0.256. *Post hoc* paired comparisons indicated no significant differences relative to the minimum values observed in the experimental conditions ($P = 0.982$ for the TA; $P = 0.310$ for the FCR). This implies that iCI simulations induced a synchronization across MNs that closely matched the experimental observations.

By contrast, simulations only considering cCI (Fig. 3*D*, curves averaged across simulations for each condition) did not lead to synchronization events at around 0 s that were comparable to the experimental data or to the simulations considering iCI. When cCI were simulated, the average minimum SPIKE distance level was always significantly greater than for experimental data, regardless the frequency of the input. Specifically, the average minimum SPIKE distance reached in each case was 0.278 for the cCI at *F*DR; 0.291 for the cCI at 20 Hz; and 0.290 for the cCI at 1 Hz. Similarly, when simulating the no-CI scenario (Fig. 3*D*, curve averaged across simulations), the average minimal SPIKE distance was 0.292. All of the minimal SPIKE distance values in these conditions were significantly larger than the minimal SPIKE distance of the experimental datasets or the simulations with iCI (*P* values and effect sizes reported in Table 2).

Overall, these results demonstrate the global nature of the observed MN synchronization by confirming

its presence across different muscles (TA and FCR), and show that it cannot be explained by the intrinsic statistical properties of the data, as evidenced by the surrogate analyses. Furthermore, they show that cCI alone, or the mere variability of MN activity in the absence of CI, cannot account for the sporadic and transient synchronization events observed experimentally. By contrast, iCI successfully reproduces the experimental findings, supporting its plausibility as an underlying input mechanism.

To further characterize the effects of iCI on MN activity at the population level, we computed and compared the PSD and IMC functions obtained from experimental data and the simulations. The PSD and IMC functions are typically used to analyze the population activity of MNs because they measure the spectral distribution of the signals that are sampled by the pool. Figure 4*A*,*B* shows the average PSD and IMC obtained from the TA. These plots are qualitatively similar to those reported previously in the literature (Farina & Holobar, 2016; Ward et al., 2013) and show two characteristic features. The PSD is dominated by a peak at the frequency matching the average *F*DR of the MNs (this is typically around 10 spikes s$^{-1}$ in the tibialis anterior muscle for low-force contractions as in this study). However, this peak is not present in the IMC, indicating that the MN pool does not contain an oscillatory common signal at *F*DR that dominates the spectra of the MNs. A similar pattern is reproduced for the FCR muscle in Fig. 4*C*,*D*, showing a PSD also governed by the peak at the average *F*DR of the pool, whereas the coherence profile lacks this peak.

The same analysis using the simulated data with iCI with a rate of 1 Hz (Fig. 4*E*,*F*) leads to results that are very similar to those observed experimentally, with a relatively low IMC level at frequencies around *F*DR despite the presence of a prominent peak at that same frequency in the PSD. This is probably related to the induced transient synchronization events of MNs at the times of the impulses. Because of the short-lived nature of these events, the IMC is not affected by them and shows a lack of common contents in the MNs at *F*DR (Fig. 4*F*).

Simulations using cCI (Fig. 4*G*,*H*) lead to PSD and IMC functions that were similar in shape to the power spectrum of the CI itself (as expected because of the capacity of MN pools to linearly amplify cCI) (Farina & Negro, 2015; Farina et al., 2014). In these cases, a prominent peak at the frequency of the CI can be observed in the IMC, which implies that in the presence of only cCI, the combined activity of MNs should tend to be dominated by contents related to the inputs. Finally, the scenario in which no CI are simulated leads to values of the IMC close to 0 at all frequencies, which is expected given the absence of common signals across the MNs (Fig. 4*J*).

In summary, we observe relevant similarities between the experimental data in both muscles and simulations

using iCI. These similarities are not reproduced when simulations do not include impulsive inputs but instead use cCI. These results strongly suggest that the MN pool receives at least part of its drive in the form of intermittent iCI. The next sections address how these impulsive inputs can affect the linearity assumptions typically made in the interpretation of signals sampled by MN pools

## Impulsive inputs alter the linearity of MN pools

Under the influence of cCI, MN pools act as linear amplifiers of these inputs (Farina et al., 2014). However, in the case where iCI also drive individual MN activity, the repeated transient synchronization across MNs (once every time an impulse is produced) may cause the linear behaviour of the system to be altered. To assess this, we conducted several tests using the simulations involving iCI.

First, we assessed how the fundamental frequency of the iCI (i.e. the average rate at which impulsive events occur) affects the power spectrum of the summed activity of MNs. In line with the previous section, 30 MNs were used to obtain the PSD of their summed activity (randomly selecting 30 different MNs several times and averaging the obtained results). If the iCI do not affect the linear behaviour of the MNs, it should be expected that the PSD of the summed MN activity is only modified according to the spectral contents of the impulsive input (a linear

transmission). However, we observe that this is not the case when iCI are simulated. Figure 5 shows the evolution of the PSD of the summed MN activity when the rate of occurrence of the impulsive inputs increases from 0 events s$^{-1}$ to 4 events s$^{-1}$. As the rate of impulsive events increases, the PSD average level increases, although, relevantly, the PSD level at *F*DR increases at a higher speed than the rest of the frequencies. Indeed, we observe that the power at *F*DR in the cumulative spike train increases monotonically when the rate of impulsive events increases despite the gain at that frequency shows a very different trend (the inset in Fig. 5 shows that the gain at *F*DR does not follow a linear increase with the input frequency). Such an outcome is inconsistent with the predictions of a purely linear system, where the output power at *F*DR would be expected to scale proportionally with the input power at the same frequency.

Because the linear behaviour of the MN pool may depend on the number of MNs considered, we conducted a second test where we compared the gain (output power divided by the input power) at *F*DR for different numbers of MNs when the CI was either an iCI producing one impulsive event per second or a cCI modelled as a pure sinusoid with a frequency *F*DR and with a power at that frequency matched to the power of the iCI (i.e. both CI had the same power at *F*DR but different power at other frequencies) (Fig. 6*A*). The gain at *F*DR of the summed activity of MNs increased with the number of MNs considered in both scenarios (Fig. 6*B*). However,

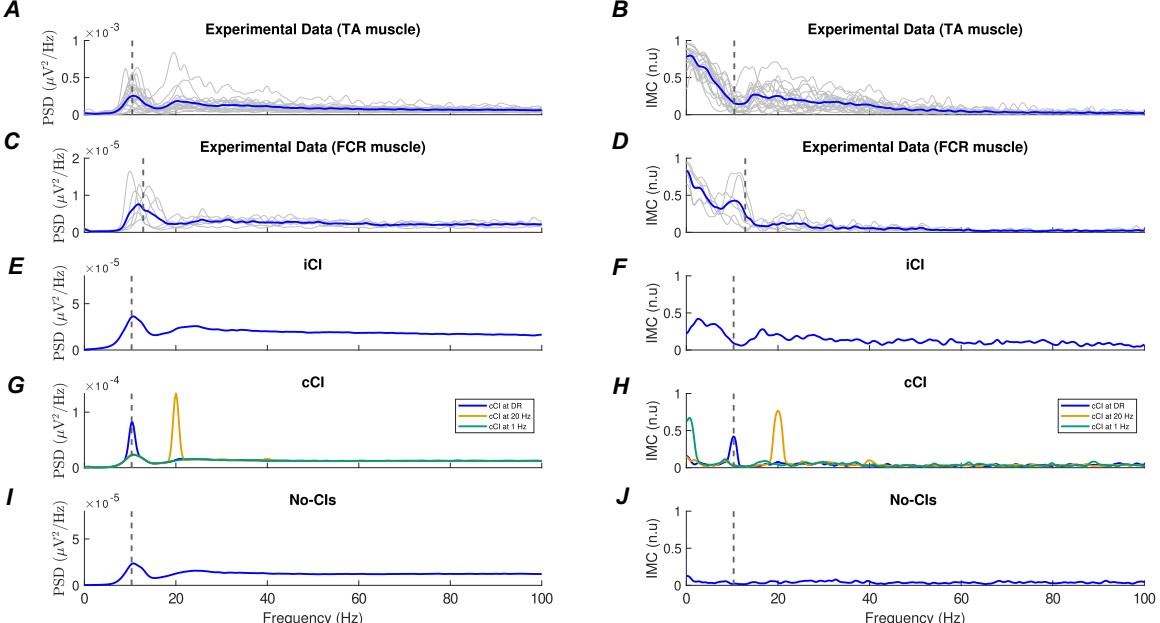

**Figure 4. PSDs and IMCs of the summed activity of MNs for experimental data**
TA (*A* and *B*) and FCR (*C* and *D*); simulations using iCI (*E* and *F*), simulations using cCI (*G* and *H*) and simulations that do not include any CI (*I* and *J*). Curves are averaged across subjects or simulations. The vertical dashed line indicates the mean discharge rate of the pool for each condition. PSD and IMC recorded for both muscles are better modelled by simulations involving iCI than by any other simulated condition.

although the power at *F*DR of the summed MN activity increased linearly in the case of the cCI, the trend became exponential with the iCI ($R^2 = 0.97$ of fit for an exponential model of the form $y = a \cdot e^{b \cdot x}$, with parameters $a = 0.60$ and $b = 0.01$, where $y$ is the gain at *F*DR and $x$ is the number of MNs included) and the output power at *F*DR diverged from the power obtained with the cCI as more MNs were considered. This reflects the deviation from the linear behaviour when the impulsive input is used (two input signals with the same power at a certain frequency lead to output signals with different power at that same frequency) and it also reflects that the non-linear effects produced by the MN pool do not decay when larger pools of MNs are considered. The spectra of the cumulative spike train for both input scenarios across the different number of MNs evaluated are shown in Fig. 6*C* (cCI) and D (iCI) respectively. In both spectrums, it can be observed that the peak at *F*DR increases more rapidly with the number of MNs in the iCI case (Fig. 6*D*).

In summary, the presence of iCI can alter the behaviour of MN pools as linear amplifiers of CI. Below, we analyze how this effect can deteriorate the estimation of cCI in the presence of iCI driving MN activity.

## Impulsive input-induced synchronization alters signal transmission by the MN pool

Here, we examine how iCI can affect the transmission of cCI by the MN pool. The main objective in this analysis is to assess how reliably cCI in the beta band or in the

low-frequency band (related to the force control) can be decoded from MN activity when the MN pool is also driven by iCI, which disrupt its linear transmission properties as previously described.

To assess this, in different sets of simulations, we evaluated the transmission of cCI at 1 Hz and 20 Hz by a MN pool in two different conditions: the MN pool receiving only cCI or receiving both cCI and iCI. We tested increasing power levels for this cCI (increasing levels of signal to noise ratio), with the maximum corresponding to the maximal power that did not significantly influence the coefficient of variation in the simulated force, as detailed in the Methods. For each power level, we tested how it was transmitted by the MN pool with and without an additional iCI, evaluating this for 10, 30 and 177 MNs in the cumulative spike train generated to estimate the input from the output of the MNs. The impulsive input was generated with a fundamental frequency of 1 Hz and the same power as in previous sections. To assess the transmission of the cCI, we measured the maximal value of the cross-correlation function between that input and the cumulative spike train of the MN pool filtered around 20 Hz or 1 Hz (depending on the frequency of the simulated cCI). For this analysis, the MN pool was firing at an average rate of ∼10.4 Hz, in line with previous sections.

Figure 7 summarizes the results obtained in these simulations. The main finding is that the presence of iCI consistently reduces the correlation between the cumulative spike train and the cCI, regardless the number of MNs included, the frequency of the cCI or the input

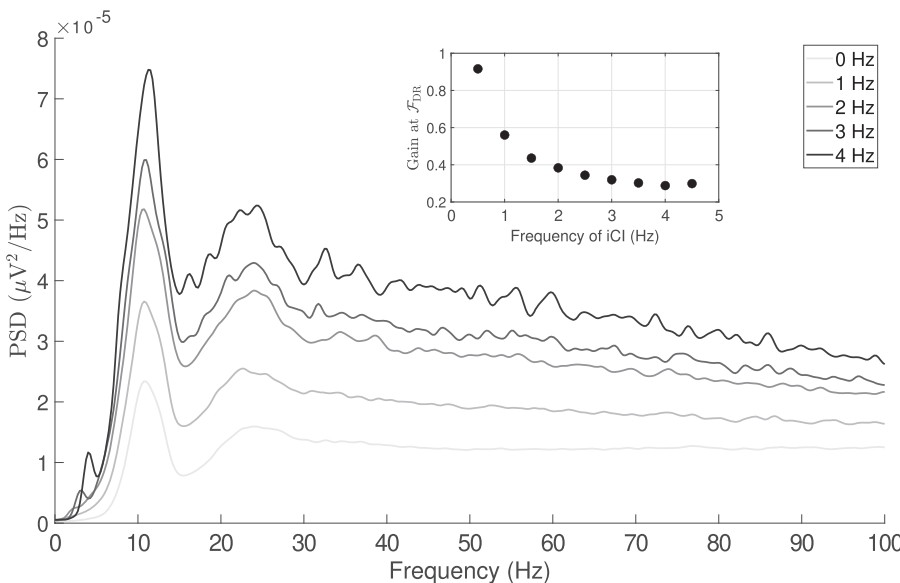

**Figure 5. Spectrum of a pool of 30 MN receiving iCI from 0 Hz (no bursts) to 4 Hz, in increments of 1 Hz**
There is a monotonous increase in power at *F*DR when the rate of occurrence of impulsive inputs increases from 0 Hz to 4 Hz. This reflects a non-linear behaviour of the MN pool, as reflected in the inset showing the ratio (gain) of the power at *F*DR between the input of the system (iCI) and the output (the sum of MN spike trains).

power level. However, this reduction in the correlation was more pronounced for lower powers of the CI (low signal to noise ratios), as expected.

Two main findings can be identified from these simulations. First, the correlation drop caused by the presence of iCI becomes more pronounced with increasing numbers of MNs (the difference between the blue and the yellow curves is more pronounced when more MNs are included). This is in line with Fig. 6*B*, which shows that increasing the number of synchronized MNs in the cumulative spike train amplifies the non-linear behaviour of the pool. Second, the correlation drop observed at 1 Hz (Fig. 7, top row) is greater than that at 20 Hz (Fig. 7, bottom row) for all MN group sizes evaluated, indicating that the disruptive effects of the iCI are more pronounced at lower frequencies.

It is important to remark that the impulsive events in these simulations appeared only once per second, whereas the correlation was computed over 30 s simulations. This indicates that the distortion caused by each pulse was brief and localized, allowing the MN pool to recover linear transmission properties during the remaining time, resulting in the high overall correlation values obtained in the simulations.

Overall, these analyses indicate that iCI impair the MN pool ability to sample and linearly transmit cCI.

This outcome is consistent with the fact that impulsive inputs push the MN pool toward a non-linear behaviour. The extent of this disruption depends on the balance between the power of the cCI and the power (or frequency) of the iCI: the weaker the cCI, the more pronounced the effect of the impulses on its transmission, and, conversely, stronger cCI are more resilient to the disruptive effects of the iCI.

## Discussion

Studying the population activity of spinal MNs is essential for understanding the neural strategies underlying muscle control and force generation. Many motor neuron models account only for continuous inputs characterized by relatively slow dynamics (Farina & Holobar, 2016; Farina et al., 2014; Watanabe & Kohn, 2017; Zicher et al., 2023). However, within this framework, certain features of experimentally observed MN activity cannot be accurately reproduced. In this study, we propose and analyze the existence of common impulsive signals transmitted to the MNs that force a temporal synchronization of their action potentials. Using both simulations and experimental data from healthy human subjects, for the first time, we indirectly showed evidence that MN pools

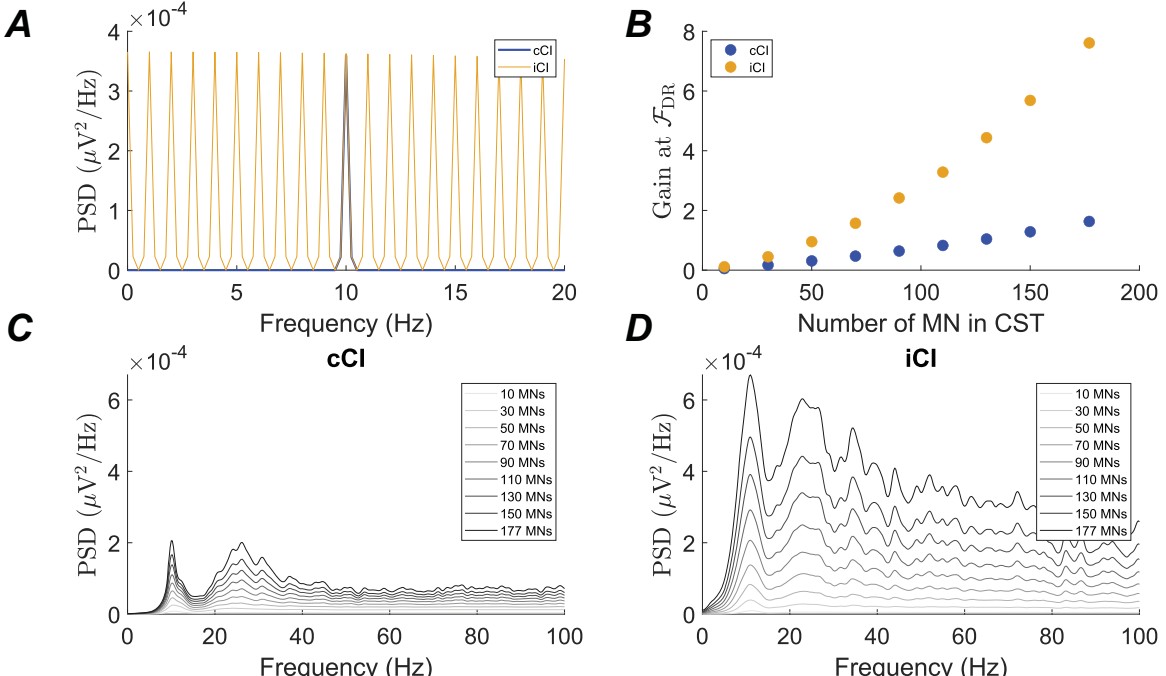

**Figure 6. cCI and iCI elicit different behaviors in the MN pool**
*A*, spectrum of two types of CI simulated as a pure tone (blue) or a burst train (yellow) with 1 Hz fundamental frequency. *B*, ratio (gain) between the output and input power at the average *F*DR of the simulated MNs when the CI is a pure tone (blue) or a burst train (yellow). Each point corresponds to the average across 10 simulations. *C*, PSD of a MN pool receiving a sinusoid at *F*DR Hz, evaluated for different number of MNs in the cumulative spike train. *D*, PSD of a MN pool receiving a iCI at a frequency of 1 Hz, evaluated for different number of MNs in the cumulative spike train.

can receive significant influences through iCI that alter their linear behaviour. In this context, we use the term impulsive implying that these neural signals that drive MN activity occur during periods of time that are much shorter than the spiking frequency of the driven MNs. Notably, evidence of iCI is observed not only in a single experimental context, but also across two distinct muscles from different limbs. This indicates that the MN synchronization mechanism we describe probably reflects a global feature rather than a local phenomenon confined to a single muscle. The presence of such iCI can successfully replicate certain aspects of the MN activity that are not explained by simulations only including continuous CI (cCI), such as the common spectral contents across MNs and the spiking synchronization at the population level. These findings have important relevance in the current understanding of motor control and also have an impact on the development of neural interfaces that extract information from the activity of

populations of MNs. Below, we discuss the potential role of these impulsive inputs and their possible origins.

## Potential role of the impulsive inputs on the motor pathway

Motor control models commonly assume that the motor cortex exerts continuous control over muscles (Shadmehr & Wise, 2004). However, it has also been proposed that higher-level motor commands may be intermittent (Karniel, 2013). One compelling idea is that the motor system decomposes complex or prolonged actions into discrete submovements that serve as building blocks for full motor behaviors. Such segmentation could arise from a limited temporal planning horizon, requiring ongoing updates during execution, as suggested by computational theories (Guigon, 2023). Although submovement-rate modulation is often associated with slow or prolonged

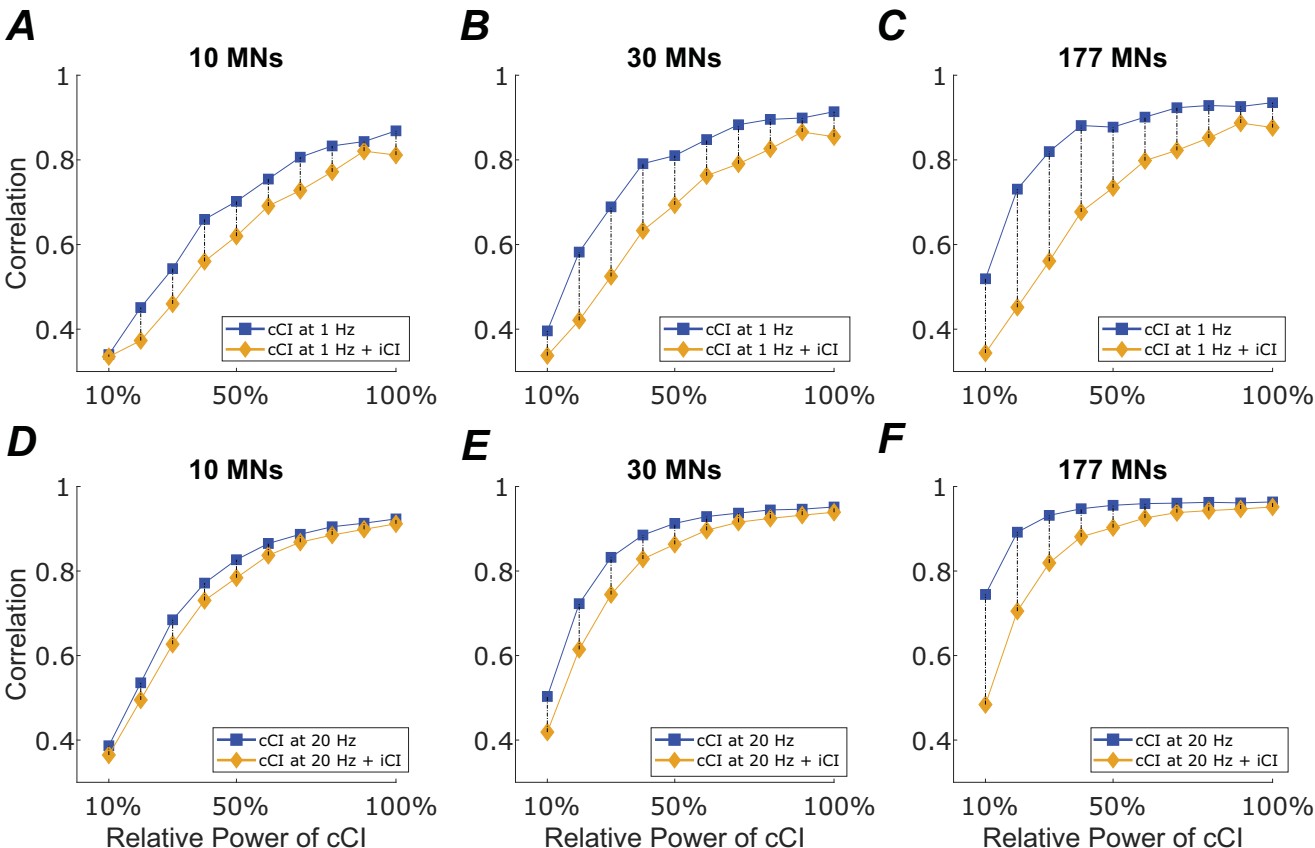

**Figure 7. iCI distort the transmission of other cCI and the relative severity of this distortion depends on the number of MNs considered and the power of the cCI relative to the power of the iCI**
The correlation between the filtered cumulative spike train and a cCI is shown at 1 Hz (*A*, *B* and *C*) or 20 Hz (*D*, *E* and *F*). The cumulative spike train was obtained summing the activity of 10 MNs (*A* and *D*), 30 MNs (*B* and *E*) or 177 MNs (*C* and *F*) and then bandpass filtering the result around 1 Hz or 20 Hz (depending on the simulated input). The black dashed lines highlight the correlation difference for each power level used for the cCI. The power of the cCI is expressed as a percentage of the maximal power used (maximal power such that it does not modify the coefficient of variation of the simulated force).

movements, it may apply to a much wider range of motor behaviors.

Co-ordinating these submovements might require intermittent control signals (e.g. impulsive bursts) that synchronize a substantial proportion of the active motor neurons in a pool. These synchronized events could act as neural 'reset' points, aligning the system for the initiation of the next submovement. Beyond preparation, such bursts might also serve a sensory–motor monitoring role, allowing the system to reassess the state of the muscles even during steady or constant tasks.

The presence of iCI could have measurable consequences for movement. Low-frequency components of the cumulative spike train strongly correlate with the force output of a muscle (Thompson et al., 2018). Given the low rate of impulsive events observed here (typically less than 1 event s$^{-1}$), their influence would be expected primarily in the low-frequency range of motor neuron activity and force output (1–2 Hz) (Christou et al., 2003, 2004; De Luca et al., 1982). Indeed, it is well established that the CNS struggles to maintain perfectly stable force (Enoka & Farina, 2021) and force fluctuations at target levels above 10% of the maximal voluntary contraction appear to be largely driven by variability in the common modulation of motor unit discharge times (Negro & Farina, 2012). In this framework, iCI could contribute to force variability by injecting power into the band at 0–2 Hz, thereby increasing signal variance and reducing steadiness compared to a system without such input.

It is also possible that impulsive inputs have no direct functional role. They may simply reflect incidental activity originating from other brain regions (e.g. the motor cortex) that reaches spinal motor neuron pools because the nervous system has not evolved mechanisms to suppress them. In such a scenario, the occurrence of iCI would be a byproduct of neural architecture rather than an adaptation. For example, synchronized bursts could arise from passive attentional processes triggered by salient external stimuli, such as a loud sound (Novembre et al., 2018), leading to transient excitations in motor cortical areas that propagate downstream to spinal motor neurons without functional filtering. Further investigation in subjects with movement disorders or brain or spinal lesions will be informative in addressing this question and determining whether fundamental differences exist in the rate or effects of impulsive events compared with healthy conditions. Such studies would allow a more accurate assessment of the potential role of this effect.

### Possible origin of the impulsive inputs

As a result of the nature of our experimental dataset and the limited access to spinal MNs, it is difficult to accurately characterize the properties of these impulsive inputs, such as their strength, periodicity or innervation pattern, among other features. Furthermore, the absence of simultaneous recordings from other regions of the nervous system makes it challenging to speculate on the potential sources of these impulsive inputs. One possible source for this activity is the motor cortex because there is evidence in the literature suggesting the existence of short-lived events that can be sent to the MN pool, such as the beta bursts (Bräcklein et al., 2022; Echeverria-Altuna et al., 2022) or drastic changes in cortex activity related to sequences of inhibition and excitation, modulated by external stimuli (Novembre et al., 2018). The reticular formation is also a structure to be considered in this context. The reticulospinal tract has been described as one of the main sources of neural drive during voluntary control of movements (Glover & Baker, 2022). Interestingly, the reticulospinal tract has also been confirmed to mediate the acceleration of reaction time when a loud sound accompanies the cue (Tapia et al., 2022). This suggests that the reticulospinal tract is able to generate additional inputs to the MN pool modulating voluntary movement, in addition to the commands controlled by the motor cortex.

Future research analyzing the particular role of these superior structures during the tracking of different forces will provide further clarity about the generation of these intermittent events.

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

## Additional information

### Data availability statement

Data from this study will be made available to qualified investigators upon reasonable request to the corresponding author. The computational model and analysis code used in this study were implemented in MATLAB. The code is available upon request from the corresponding author.

### Competing interests

The authors declare that they have no competing interests.

### Author contributions

The experiments of the TA dataset were conducted at University College London, and the experiments of the FCR dataset were conducted at the University of Zaragoza. The simulations were carried out at the University of Zaragoza. All authors approved the final version of the manuscript submitted for publication. All authors agree to be accountable for all aspects of the work. All authors qualify for authorship, and all those who qualify for authorship are listed. J.Y.-M., A.P.-V., S.N.B., P.L., D.F. and J.I. were responsible for conceptualization. J.Y.-M., A.P.-V. and J.I. were responsible for investigations. J.Y.-M. and S.N.B. was responsible for software. J.Y.-M., A.P.-V., S.N.B., P.L., D.F. and J.I. were responsible for analysis. J.Y.-M., A.P.-V. and J.I. were responsible for visualization. J.Y.-M., A.P.-V., S.N.B., P.L., D.F. and J.I. were responsible for writing the manuscript. J.Y.-M., A.P.-V. and J.I. were responsible for FCR data acquisition. J.I. was responsible for TA data acquisition and funding acquisition.

### Funding

JY-M, PL and JIP were supported by the European Research Council (ERC) under the European Union's Horizon Europe research and innovation program (ECHOES project; ID – 101077693). JY-M received a grant from Gobierno de Aragón (ORDEN EMC/590/2025). JIP was supported by MICIU/AEI and FEDER, UE (Grant PID2022-138585OA-C32) and by a consolidación investigadora grant (CNS2022-135366) funded by MCIN/AEI/10.13039/501100011033 and UE's nextGenerationeU/PRTR funds. AP-V was supported by the European Union's Horizon Europe research and innovation programme under the Marie Skłodowska-Curie grant agreement No. 101151398. Computations were performed using ICTS NANBIOSIS (HPC Unit at University of Zaragoza). DF was supported by the EPSRC under the project NISNEM Technology (EP/T020970/1) and by the ERC under the Synergy Grant project Natural BionicS. D.F and S.N.B were partly funded by BBSRC grant BB/V00896X/1.

### Keywords

impulsive common input, motor control, motor neurons, movement, non-linear behaviour

## Supporting information

Additional supporting information can be found online in the Supporting Information section at the end of the HTML view of the article. Supporting information files available:

**Peer Review History**

