## [Peer Review History · The Journal of Physiology]

Spinal Motor Neuron Pools May be Partly Driven by Impulsive Common Inputs

Javier Yanguas, Alejandro Pascual Valdunciel, Stuart N Baker, Pablo Laguna, Dario Farina, and Jaime Ibanez Pereda
DOI: 10.1113/JP290395

Corresponding author(s): Jaime Pereda (jibanez@unizar.es)

Review Timeline:

Submission Date:	25-Oct-2025
Editorial Decision:	22-Jan-2026
Revision Received:	04-Mar-2026
Accepted:	13-Apr-2026

Senior Editor: Richard Carson

Reviewing Editor: Mathew Piasecki

Transaction Report:

Re: JP-RP-2025-290395 "Spinal Motor Neuron Pools May be Partly Driven by Impulsive Common Inputs" by Javier Yanguas, Alejandro Pascual Valdunciel, Stuart N Baker, Pablo Laguna, Dario Farina, and Jaime Ibanez Pereda

Dear Dr Pereda,

Thank you for submitting your manuscript to The Journal of Physiology. It has been assessed by a Reviewing Editor and by 2 expert referees and we are pleased to tell you that it is potentially acceptable for publication following satisfactory major revision.

Please address all the points raised and incorporate all requested revisions or explain in your Response to Referees why a change has not been made. We hope you will find the comments helpful and that you will be able to return your revised manuscript within 2 months. If your article is NOT for a Special Issue, you may have 9 months to revise. If you require an extension, please contact journal staff: jp@physoc.org. Please note that this letter does not constitute a guarantee for acceptance of your revised manuscript.

REVISION CHECKLIST:

We look forward to receiving your revised submission.

Yours sincerely,

Richard Carson
Senior Editor
The Journal of Physiology

REQUIRED ITEMS

- 1) - Author photo and profile. First or joint first authors are asked to provide a short biography (no more than 100 words for one author or 150 words in total for joint first authors) and a portrait photograph. These should be uploaded and clearly labelled together in a Word document with the revised version of the manuscript. See Information for Authors for further details.
- 2) - The reference list must be in alphabetical order, rather than numbered, to comply with our Journal format.
- 3) - Please upload separate high-quality figure files via the submission form.
- 4) - Please ensure that any tables are editable and in Word format, and wherever possible, embedded in the article file itself.
- 5) - Please ensure that the Article File you upload is a Word file.
- 6) - Papers must comply with the Statistics Policy: https://jp.msubmit.net/cgi-bin/main.plex?form_type=display_requirements#statistics.

In summary:

- If $n \leq 30$, all data points must be plotted in the figure in a way that reveals their range and distribution. A bar graph with data points overlaid, a box and whisker plot or a violin plot (preferably with data points included) are acceptable formats.
- If $n > 30$, then the entire raw dataset must be made available either as supporting information, or hosted on a not-for-profit repository, e.g. FigShare, with access details provided in the manuscript.
- 'n' clearly defined (e.g. x cells from y slices in z animals) in the Methods. Authors should be mindful of pseudoreplication.
- All relevant 'n' values must be clearly stated in the main text, figures and tables.
- The most appropriate summary statistic (e.g. mean or median and standard deviation) must be used. Standard Error of the Mean (SEM) alone is not permitted.
- Exact p values must be stated. Authors must not use 'greater than' or 'less than'. Exact p values must be stated to three

significant figures even when 'no statistical significance' is claimed.

7) - Please include an Abstract Figure file and an Abstract Figure legend. An appropriate figure legend, which should not exceed 150 words in length, should be included in the main manuscript file. The Abstract Figure is a piece of artwork designed to give readers an immediate understanding of the research and should summarise the main conclusions. If possible, the image should be easily 'readable' from left to right or top to bottom. It should show the physiological relevance of the manuscript so readers can assess the importance and content of its findings. Abstract Figures should not merely recapitulate other figures in the manuscript. Please try to keep the diagram as simple as possible and without superfluous information that may distract from the main conclusion(s). Abstract Figures must be provided by authors no later than the revised manuscript stage and should be uploaded as a separate file during online submission labelled as File Type 'Abstract Figure'. Please also ensure that you include the figure legend in the main article file. All Abstract Figures should be created using BioRender. Authors should use The Journal's premium BioRender account to export high-resolution images. Details on how to use and access the premium account are included as part of this email.

8) - Please include a full title page as part of your main article (Word) file, which should contain the following: title, authors, affiliations, corresponding author name and contact details, keywords, and running title.

9) - Please ensure that all figures and tables have a title and legend, and that they have been cited within the main article text.

EDITOR COMMENTS

Reviewing Editor:

Your manuscript has been reviewed by those with appropriate expertise, and although several positive points are noted there are significant concerns that must be addressed before publication can be considered. I draw particular attention to the use of a single recording from the human data, as outlined by reviewer#1; any questions around the repeatability of these outcomes could well be answered with your existing data, and I encourage the authors to pursue this. Please also consider strengthening the rationale and the importance of the findings in the introduction and discussion, respectively, as highlighted by reviewer#2.

Senior Editor:

As highlighted by the reviewers and by the Reviewing Editor, the theoretical significance of the work and the quality of the computational modeling and simulation is recognised. In view of the potential impact of the conclusions therefore, it is essential that the generality of the findings can be demonstrated. In this regard therefore, and to reemphasise the points that have been made by the Reviewing Editor and referees, further consideration of the submission will require that experimental evidence in support of the central proposition be obtained for a broader representation of recordings, not only from the TA muscle (i.e., in the preparation considered in the original submission) but also for other muscles, and ideally in other task contexts. The point has also already been made that the section of only a small number of recordings from all those available for the preparation described, is a matter of concern.

In short, it should be demonstrated that the phenomenon under consideration is manifested more widely, i.e., beyond the specific context considered here.

REFEREE COMMENTS

Referee #1:

I thank for the opportunity to review the manuscript entitled "Spinal Motor Neuron Pools May be Partly Driven by Impulsive Common Inputs". The study addresses whether motor neuron pools may receive, in addition to continuous common inputs, brief impulsive inputs that transiently synchronize motor unit discharge and alter the linear transmission properties of the pool.

General comment:

The computational modeling and simulation framework is carefully developed and represents the strongest aspect of the manuscript. The simulations demonstrate that impulsive common inputs could, in principle, generate transient synchronization events and modify the spectral properties of pooled motor neuron activity. However, the experimental data, as currently presented, does not provide sufficiently robust support for the central claims advanced in the Discussion. In particular, given the novelty and potential impact of the proposed mechanism, it is essential to demonstrate that the reported synchronization phenomena are consistently observed across trials, and not dependent on a single selected recording per subject. At present, the experimental analysis relies heavily on one chosen trial, which raises concerns regarding the reliability and generalizability of the findings. Without clear evidence that these effects are reproducible either across repeated trials or, ideally, across different muscles, the experimental results remain preliminary and do not fully substantiate the strong mechanistic conclusions drawn from the simulations.

More generally, several methodological choices are insufficiently justified, giving the impression that some parameters may have been selected post hoc rather than being grounded in prior literature or supported by explicit statistical reasoning.

Specific comments

1. Lines 96-99: The justification provided for selecting only one recording per subject is not entirely convincing. The tibialis anterior is among the most extensively studied muscles in the motor unit literature precisely because motor units can often be reliably decomposed and, in many cases, tracked across sessions. Given the minimum inclusion criterion of 15 motor units, it seems plausible that at least a subset of units could have been followed across the two available trials. More importantly, the main concern is that all experimental conclusions are drawn from a single chosen trial per subject. How robust are the reported results? It is critical to show that these population-level synchronization phenomena, claimed to reflect impulsive common inputs, are consistently observed when analyzing the other ramp-and-hold contraction, even if exact unit tracking across trials is not feasible.
2. The rationale for defining a sufficiently large population as 15 motor units is unclear. This threshold should either be justified quantitatively (e.g., through sensitivity analyses or statistical considerations) or supported by prior literature. As it stands, it appears that this criterion may have been used primarily to exclude 5 of the 19 subjects in the original dataset. For transparency, the authors should report how many motor units were decomposed in each of the 19 subjects and clarify how sensitive the results are to this selection threshold.
3. Lines 220-221: the manuscript states that a 30-second segment with constant force was selected to ensure steady motor unit firing. How exactly was this period identified? Was it based on minimizing the coefficient of variation of force? If so, this does not necessarily guarantee that all decomposed motor units were firing continuously and stably during that interval. Please clarify whether motor unit firing behavior was explicitly inspected to confirm this assumption.
4. Lines 230-232: the exclusion of motor units with higher discharge rate variability requires stronger justification. From a mechanistic standpoint, it would actually be highly informative to test whether impulsive common inputs also affect units with more variable firing behavior. The current exclusion criterion may bias the analysis toward more regular units and should either be better justified or complemented by additional analyses including these excluded units.
5. From what I understood, several two-sample t tests were used. The use of multiple two-sample t-tests to compare conditions is not ideal for the number of comparisons performed and it is not correct statistically. A one-way ANOVA, for instance, would provide a more statistically sound framework, followed by post hoc tests where appropriate. The authors should justify their current approach or revise the statistical analysis accordingly. I also suggest the inclusion of effect sizes when appropriate.
6. In Figure 2, there are intervals outside the gray highlighted windows where apparent motor unit synchronization occurs without a corresponding increase in the amplitude of the summed motor neuron activity at the FDR. For example, in Figure

2B, the interval between the first two gray rectangles illustrates this discrepancy. This observation seems to contradict the stated relationship between synchronization events and FDR amplitude increases and should be addressed explicitly.

Referee #2:

Summary

The primary aim of this study was to determine whether the activity of the motor neuron pool is driven at least in part by impulsive-like input components (iCI). The authors combined analysis of experimental data from human subjects with three simulated scenarios designed to reproduce experimental observations. Experimental data show high synchronization at the population level, consistent with simulations testing the hypothesis that motor neuron pools receive iCI. The study also demonstrates that iCI alter the linear behavior of motor neuron pools and affect estimation of continuous common inputs (cCI) based on motor unit activity.

Overall, this manuscript represents a solid study and an important advancement in understanding neural input to motor neuron pools. The techniques implemented are appropriate and well executed, and the authors are well equipped to perform this research. The methodology is detailed and sound.

Suggestions:

Abstract and Intro

The abstract and introduction should include a clearer and earlier link to spinal inputs and spinal circuitry. Practical implications such as neural interfaces and motor control models should be highlighted earlier. The authors should also simplify technical jargon in the abstract, for example by briefly explaining what is meant by linear behavior and why it matters. The introduction should also include a short paragraph summarizing limitations of previous studies to justify the need for this work. Pulling these points into the abstract and introduction will broaden readership.

Methods

Subject Selection: The authors state that the experimental data includes only two female subjects out of nineteen. The authors should clarify whether additional female subject data were excluded, whether differences were observed compared to other subjects, and whether any sex-related differences were expected.

Statistics: While p-values are reported, effect sizes or confidence intervals would provide better insight into practical significance.

Figures

Figure legends should explain the significance of the data, not only the technical details.

Discussion

The discussion begins to link this work to neural commands and motor control. This section should expand on functional consequences, such as how these findings might influence motor learning, force steadiness, and rehabilitation strategies.

END OF COMMENTS

Dear Editors and Reviewers,

We sincerely thank you for your positive response and for considering our manuscript of interest. We greatly appreciate the constructive comments provided by the reviewers, which we believe have helped us to improve the quality and clarity of our work.

In this letter, we address the reviewers' comments on a point-by-point basis, in accordance with the instructions provided. We hope that the revised manuscript meets the high standards of excellence required for publication in the *Journal of Physiology*.

Senior editor

As highlighted by the reviewers and by the Reviewing Editor, the theoretical significance of the work and the quality of the computational modeling and simulation is recognised. In view of the potential impact of the conclusions therefore, it is essential that the generality of the findings can be demonstrated. In this regard therefore, and to reemphasise the points that have been made by the Reviewing Editor and referees, further consideration of the submission will require that experimental evidence in support of the central proposition be obtained for a broader representation of recordings, not only from the TA muscle (i.e., in the preparation considered in the original submission) but also for other muscles, and ideally in other task contexts. The point has also already been made that the section of only a small number of recordings from all those available for the preparation described, is a matter of concern.

In short, it should be demonstrated that the phenomenon under consideration is manifested more widely, i.e., beyond the specific context considered here.

We agree with the suggestion and thank the Editor for the constructive feedback. As indicated in our response to reviewer 1, to demonstrate that the observed phenomenon is not limited to an isolated experimental context, we have expanded the validation of our results. Specifically, we have also analyzed the previously excluded TA block. As a reminder, blocks were originally excluded solely on the basis of the number of decomposed motor units—a criterion that does not introduce bias in our analyses or conclusions. The results obtained were equivalent to the results of the included block and led to the same conclusions. A figure comparing both blocks is given in this response letter.

Importantly, we have also included an additional dataset from a different muscle in the upper limb (flexor carpi radialis) recorded using intramuscular high-density EMG. To do this, we had to use recently developed multielectrode arrays, since otherwise it is difficult to track sufficiently large sets of motor units from upper limb muscles. Overall, the results obtained from the different sets of experimental data remain largely stable and comparable among them. We believe that these

additions provide sufficient evidence supporting the phenomenon under investigation.

Regarding the suggestion to include additional task contexts, several considerations should be noted. First, most studies investigating motor units and neural inputs to muscles rely on recordings obtained during isometric contractions. This is because motor unit decomposition can be performed reliably under these conditions, whereas it becomes substantially less reliable during dynamic activities due to current limitations of decomposition algorithms.

Exploring alternative task contexts would also require formulating specific hypotheses about the functional role of impulsive inputs. We believe this lies beyond the scope of the present study, whose primary objective is to provide the first evidence for the existence of impulsive neural signals in the peripheral nervous system.

Importantly, the proposed framework remains highly relevant even when demonstrated solely during sustained contractions. First, we provide evidence for a distinct type of input to motor units that critically shapes the spectral properties of motor unit pools. This finding has major implications for the interpretation of results in all studies based on motor unit pool analyses. Second—and perhaps more importantly—we present evidence of a previously undescribed signal propagating along the motor nervous system. This result raises fundamental questions about communication between the central nervous system and muscles and offers new insight into the extent to which motor units receive inputs not directly related to the requirements of the motor task.

Reviewer 1

General Comments:

The computational modelling and simulation framework is carefully developed and represents the strongest aspect of the manuscript. The simulations demonstrate that impulsive common inputs could, in principle, generate transient synchronization events and modify the spectral properties of pooled motor neuron activity. However, the experimental data, as currently presented, does not provide sufficiently robust support for the central claims advanced in the Discussion. In particular, given the novelty and potential impact of the proposed mechanism, it is essential to demonstrate that the reported synchronization phenomena are consistently observed across trials, and not dependent on a single selected recording per subject. At present, the experimental analysis relies heavily on one chosen trial, which raises concerns regarding the reliability and generalizability of the findings. Without clear evidence that these effects are reproducible either across repeated trials or, ideally, across different muscles, the experimental results remain preliminary and do not fully substantiate the strong mechanistic conclusions drawn from the

simulations.

We thank the reviewer for the valuable feedback and for the positive assessment of our proposed new framework to model motor unit inputs. We agree that, given the potential relevance of our findings for the study of motor unit physiology, it is important to provide additional evidence demonstrating the robustness of the reported effects.

As explained below, in the response to the first specific comment of the reviewer, we have analyzed the originally discarded blocks. The results with these blocks (shown in Figure 1 below) closely match the main findings derived from the blocks with the largest number of motor neurons (MNs) decomposed, thereby supporting the consistency of the observed phenomena across blocks.

More importantly, we have also included new results with a different muscle (from the upper limb) to highlight the robustness and generalizability of our results.

Finally, we have lowered the required number of MNs to 10 units based on previous publications. Based on this criterion, we did not discard any subjects.

More generally, several methodological choices are insufficiently justified, giving the impression that some parameters may have been selected post hoc rather than being grounded in prior literature or supported by explicit statistical reasoning.

Throughout the manuscript, we have justified the selection of analysis parameters. In most cases, these choices were made to maximize the number of valid MNs included in the analysis. This approach is necessary because our main outcome measures require sufficiently large pools of MNs operating within the dynamic range of the recruitment curve.

Importantly, we avoided any analytical decisions that could bias the resulting firing synchronization indices or spectral estimates across MNs (the main outcome measures in our study). We have revised the manuscript to ensure that, whenever possible, parameter selection is supported by appropriate literature references.

Specific comments:

Lines 96-99: The justification provided for selecting only one recording per subject is not entirely convincing. The tibialis anterior is among the most extensively studied muscles in the motor unit literature precisely because motor units can often be reliably decomposed and, in many cases, tracked across sessions. Given the minimum inclusion criterion of 15 motor units, it seems plausible that at least a subset of units could have been followed across the two available trials. More importantly, the main concern is that all experimental conclusions are drawn from a single chosen trial per

subject. How robust are the reported results? It is critical to show that these population-level synchronization phenomena, claimed to reflect impulsive common inputs, are consistently observed when analyzing the other ramp-and-hold contraction, even if exact unit tracking across trials is not feasible.

Unfortunately, we could not consistently track large sets of MNs across contraction blocks (in our experience, reliably tracking 10 or more MNs is challenging in most experimental subjects). Since the reliability of our main outcome measures depends on the number of MNs considered, we decided not to merge blocks and, instead, to analyse them separately. In the article, we only considered, for each subject, the block from which more MNs could be decomposed. It is worth noting that this decision does not compromise the validity of the results obtained as it only improves the measures from population activity. Nevertheless, to address the reviewer’s comment regarding the robustness of our findings across blocks, we have also analyzed the initially discarded blocks for each participant. The results from these additional analyses, presented in this response letter, are nearly identical to those originally included in the study and thus lead to identical conclusions (see Fig. 1). They have not been included in the main manuscript since we consider them to be redundant (we do mention the fact that results with the different blocks are equivalent in the manuscript in lines 133-142). However, we would be happy to include them in the manuscript if the reviewer or the editor consider this to be critical (we originally planned to include this set of results as supplementary material, but the journal does not allow such kind of contents as supplementary information).

Figure 1 - Comparison of results between the blocks used in the manuscript and the non-considered blocks. The left panels show the PSD, IMC and SPIKE distance obtained with the blocks of TA contractions that are used for the main results. The right panels represent the same information but, in this case, using the set of the non-included blocks. Vertical dashed lines indicate mean discharge rate across subjects. Blue lines represent the average across subjects, grey lines indicate individual subjects, and yellow lines represent the average of the surrogate data.

Moreover, to further strengthen the general relevance of our findings, we have added results from recordings obtained from a different muscle (Flexor Carpi Radialis, FCR), acquired using high-density intramuscular EMG. By including data from an anatomically distinct muscle (and limb), we aim to demonstrate that the observed effects are not limited to a specific muscle. The same analyses were performed as with the TA data (PSD, IMC and SPIKE distance). The corresponding paragraphs in the results section have been modified accordingly. It is worth noting that we had to use intramuscular multielectrode arrays for this experiment since, otherwise, it would have been very difficult to decompose and track sufficiently large and reliable numbers of motor units from a single upper-limb muscle.

All the added results fully align with the original results of the study.

In short, the following changes have been included in the manuscript in line with this comment:

- In the Methods section: Pages 6-7, L128-142; L156-169; Page 13, L315-319.
- In the Results section: Changes throughout the first subsection; Figures 3 and 4 have also been updated.
- In the Discussion section: Page 27, L698-700.

The rationale for defining a sufficiently large population as 15 motor units is unclear. This threshold should either be justified quantitatively (e.g., through sensitivity analyses or statistical considerations) or supported by prior literature. As it stands, it appears that this criterion may have been used primarily to exclude 5 of the 19 subjects in the original dataset. For transparency, the authors should report how many motor units were decomposed in each of the 19 subjects and clarify how sensitive the results are to this selection threshold.

We thank the reviewer for this pertinent comment regarding the size of the analyzed MN population. In our original approach, we defined a minimum number of MNs per block to ensure robustness of the different outcome measures used. It is worth noting that a sufficiently large set of MNs is desirable in our analysis since population metrics like the intramuscular coherence or the SPIKE distance are dependent on the number of MNs considered. When the number of MNs is small, the resulting estimates become more variable (noisy), reflecting an incomplete representation of the underlying population activity. However, we believe that the criterion of only considering blocks with >15 MNs can be excessively restrictive. To adopt a less restrictive and better-justified criterion, we have now lowered the limit to 10 MN based on relevant previous publications (Farina, 2014). Accordingly, we have revised our methodology to apply a threshold of 10 MN for both the TA and FCR muscles. Since we could reliably decompose 10 or more MNs in all cases, no subjects were discarded in the new version.

We report in the manuscript the minimum and maximum number of decomposed and retained MNs with the FCR and the TA (Page 17, L458-462).

Lines 220-221: the manuscript states that a 30-second segment with constant force was selected to ensure steady motor unit firing. How exactly was this period identified? Was it based on minimizing the coefficient of variation of force? If so, this does not necessarily guarantee that all decomposed motor units were firing continuously and stably during that interval. Please clarify whether motor unit firing behavior was explicitly inspected to confirm this assumption.

The 30-s segments were selected to ensure that the highest possible number of MNs were active within the analyzed interval.

Furthermore, as described in the Methods section, we retained only those MNs a coefficient of variation of the discharge rate below 0.3, consistent with criteria adopted in previous studies now cited here (Page 13, L320-328) (Pascual-Valdunciel A. a., 2025). This criterion ensures that the considered MNs are operating within the dynamic range of their recruitment curves (fully recruited units), while excluding MNs that are intermittently recruited and de-recruited over time.

Lines 230-232: the exclusion of motor units with higher discharge rate variability requires stronger justification. From a mechanistic standpoint, it would actually be highly informative to test whether impulsive common inputs also affect units with more variable firing behavior. The current exclusion criterion may bias the analysis toward more regular units and should either be better justified or complemented by additional analyses including these excluded units.

We thank the reviewer for this valuable observation. Importantly, MNs were not excluded to remove those with intrinsically higher firing variability (we agree with the reviewer that such units would be particularly informative), but rather to ensure that the analyzed MNs were working in the dynamic range of their recruitment curves. Specifically, the coefficient of variation criterion was aimed to exclude MNs that were intermittently recruited and de-recruited throughout the recording, while retaining MNs that fired continuously and were stably identified across the analysis window. This criterion was particularly important because one of the metrics used (the SPIKE distance) depends on the inter-spike intervals of the analyzed MNs. Consequently, excluding MNs being recruited and de-recruited repeatedly was necessary to minimize artifactual distortion and noise in the resulting SPIKE distance estimates. We recognize that this rationale was not sufficiently clarified in the original manuscript, and we have therefore revised the corresponding section to make this point explicit (Page 13, L320-328).

From what I understood, several two-sample t tests were used. The use of multiple two-sample t-tests to compare conditions is not ideal for the number of comparisons performed and it is not correct statistically. A one-way ANOVA, for instance, would provide a more statistically sound framework, followed by post hoc tests where appropriate. The authors should justify their current approach or revise the statistical analysis accordingly. I also suggest the inclusion of effect sizes when appropriate.

We have revised the statistical analysis of our data according to the reviewers' comment. Because the assumption of homoscedasticity was violated, a classical one-way ANOVA was inappropriate. Therefore, we used Welch's one-way ANOVA, which is robust to heterogeneity of variances across groups. Post hoc pairwise comparisons were subsequently conducted using the Games-Howell procedure, which likewise does not assume equal variances. The relevant changes are reported in Pages 16-17, L440-449. This analysis was performed in R as MATLAB does not incorporate this functionality.

In addition, following the reviewer's comment, we have included effect sizes quantified by Cohen's d when considered appropriate. The effect sizes support the pattern of statistical significance observed. It should be noted that the reduced variability characteristic of computational simulations, compared to empirical measurements, results in larger effect sizes; therefore, these values primarily serve to confirm the robustness of the differences rather than for direct comparison with effect sizes from purely experimental studies

In Figure 2, there are intervals outside the gray highlighted windows where apparent motor unit synchronization occurs without a corresponding increase in the amplitude of the summed motor neuron activity at the FDR. For example, in Figure 2B, the interval between the first two gray rectangles illustrates this discrepancy. This observation seems to contradict the stated relationship between synchronization events and FDR amplitude increases and should be addressed explicitly.

Please, note that while instantaneous amplitude is indeed a reasonable metric for detecting synchronization events, our intention was not to suggest that it captures every individual synchronization event present in the recording, but rather that the synchronization events it identifies are reliable and appropriately characterized. Future work may be directed towards developing more refined ways to detect synchronization events from spiking activity of motor unit pools. It is also important to note that the instantaneous amplitude at the discharge rate frequency does not capture isolated or sporadic alignments of firings unless they are sustained for at least one additional discharge, since a minimum of two successive events is required to constitute an oscillatory component at that frequency.

Reviewer 2

Summary

The primary aim of this study was to determine whether the activity of the motor neuron pool is driven at least in part by impulsive-like input components (iCI). The authors combined analysis of experimental data from human subjects with three simulated scenarios designed to reproduce experimental observations. Experimental data show high synchronization at the population level, consistent with simulations testing the hypothesis that motor neuron pools receive iCI. The study also demonstrates that iCI alter the linear behavior of motor neuron pools and affect estimation of continuous common inputs (cCI) based on motor unit activity.

Overall, this manuscript represents a solid study and an important advancement in understanding neural input to motor neuron pools. The techniques implemented are appropriate and well executed, and the authors are well equipped to perform this research. The methodology is detailed and sound.

We thank the reviewer for the encouraging feedback. We have addressed each suggestion in detail in the point-by-point responses below.

Suggestions

Abstract and Intro

The abstract and introduction should include a clearer and earlier link to spinal inputs and spinal circuitry. Practical implications such as neural interfaces and motor control models should be highlighted earlier. The authors should also simplify technical jargon in the abstract, for example by briefly explaining what is meant by linear behavior and why it matters. The introduction should also include a short paragraph summarizing limitations of previous studies to justify the need for this work. Pulling these points into the abstract and introduction will broaden readership.

We thank the reviewer for the meaningful insight regarding our abstract and introduction. Overall, we have tried to make the article as accessible as possible by limiting and explaining the technical parts. Specifically in relation to the reviewer's comment, we have clarified the meaning of linear behavior in the abstract:

Page 4, L48-L50: *“This implies that the frequency content of descending and spinal oscillatory signals is preserved and faithfully transmitted to the muscles, thus, the spectral content at the output of the motor neuron pool corresponds to that of the cCI.”*

We have also added a paragraph to the introduction that better summarizes the limitations of previous studies and highlights the implications of our findings for neural interface development:

Page 6, L109-L113: *“This gap has resulted in exclusive focus on cCI under stationary conditions, while the possibility that motor neurons receive non-stationary inputs, and how they respond to such inputs, remains uninvestigated. Characterizing motor neuron responses to non-stationary inputs will have important implications for neural interfaces that decode motor intent from spinal motor neuron activity, as it provides a more complete understanding of population level behavior”*

We believe these revisions effectively address the points raised and will broaden the appeal of the manuscript to a wider readership.

Methods

Subject Selection: The authors state that the experimental data includes only two female subjects out of nineteen. The authors should clarify whether additional female subject data were excluded, whether differences were observed compared to other subjects, and whether any sex-related differences were expected.

Our current dataset does not allow us to perform a robust quantitative analysis of sex-based differences due to the technical challenges associated with decomposing a large number of motor units in female participants, such as subcutaneous tissue thickness and composition, motor unit spatial distribution and muscle anatomical cross-sectional area (Lulic-Kuryllo, 2022). We would like to note in this regard that the primary aim of this manuscript is to describe a novel way of thinking about neural inputs to muscles. We hope that future studies with larger, more balanced cohorts will be able to address relevant questions regarding possible sex differences. Regarding the exclusion of data subjects, all participants in the primary block analyzed in the study satisfied the inclusion criterion of having at least 10 MNs.

Statistics: While p-values are reported, effect sizes or confidence intervals would provide better insight into practical significance.

Effect sizes (quantified by Cohen’s d) have been included to provide better insight into the described practical significance. This is reported in Table 2 of the manuscript and in the main text.

Figures

Figure legends should explain the significance of the data, not only the technical details.

Thank you for bringing the insufficient detail in the figure captions to our attention. When considered appropriate, we have now included additional sentences to clarify the significance of the data in the captions.

Discussion

The discussion begins to link this work to neural commands and motor control. This section should expand on functional consequences, such as how these findings might influence motor learning, force steadiness, and rehabilitation strategies.

The potential influence of impulsive inputs on force steadiness is discussed in the manuscript (Page 28, L724-732). While the potential impact of impulsive inputs on rehabilitation is not straightforward (since our characterization was performed in healthy subjects without motor pathologies), we agree that it would be an interesting question to investigate this phenomenon in individuals with motor disorders, particularly to assess possible differences in the rate or effects of these inputs under altered conditions. Therefore, we have included the following information:

Page 28, L739-742: *“Further investigation in subjects with movement disorders or brain or spinal cord lesions will be informative in addressing this question and determining whether fundamental differences exist in the rate or effects of impulsive events compared with healthy conditions. Such studies would allow a more accurate assessment of the potential role of this effect.”*

Bibliography

- Farina, D. a. (2014). The effective neural drive to muscles is the common synaptic input to motor neurons. *The Journal of physiology*, 3427--3441.
- Lulic-Kuryllo, T. a. (2022). Sex differences in motor unit behaviour: A review. *Journal of Electromyography and Kinesiology*, 102689.
- Pascual-Valdunciel, A. a. (2025). Personalized mapping of inhibitory spinal cord circuits in humans via noninvasive neural decoding and in silico modeling. *Science Advances*, eadz5524.

Dear Dr Pereda,

Re: JP-RP-2026-290395R1 "Spinal Motor Neuron Pools May be Partly Driven by Impulsive Common Inputs" by Javier Yanguas, Alejandro Pascual Valdunciel, Stuart N Baker, Pablo Laguna, Dario Farina, and Jaime Ibanez Pereda

We are pleased to tell you that your paper has been accepted for publication in The Journal of Physiology.
- The reference list must be in alphabetical order, rather than numbered, to comply with our Journal format.

Yours sincerely,

Richard Carson
Senior Editor
The Journal of Physiology

IMPORTANT POINTS TO NOTE FOLLOWING ACCEPTANCE OF YOUR PAPER:

- **IMPORTANT NOTICE ABOUT OPEN ACCESS:** To assist authors whose funding agencies mandate immediate public access to published research findings, The Journal of Physiology allows authors to pay an Open Access (OA) fee to have their papers made freely available immediately on publication.

The Corresponding Author will receive an email from Wiley with details on how to register or log in to Wiley Authors where you will be able to place an order.

- You can check if your funder or institution has a Wiley Open Access Account here:
<https://authors.wiley.com/author-resources/Journal-Authors/open-access/author-compliance-tool.html>

- You can help your research get the attention it deserves! Check out Wiley's free Promotion Guide for best-practice recommendations for promoting your work at: www.wileyauthors.com/eeo/guide. You can learn more about Wiley Editing Services which offers professional video, design, and writing services to create shareable video abstracts, infographics, conference posters, lay summaries, and research news stories for your research at: www.wileyauthors.com/eeo/promotion.

- If you would like to receive our 'Research Roundup', a monthly newsletter highlighting the cutting-edge research published in The Physiological Society's family of journals (The Journal of Physiology, Experimental Physiology, Physiological Reports, The Journal of Nutritional Physiology and The Journal of Precision Medicine: Health and Disease), please click this link, fill in your name and email address and select 'Research Roundup':
<https://www.physoc.org/journals-and-media/membernews>

EDITOR COMMENTS

Reviewing Editor:

Thank you for addressing all reviewer comments.

REFeree COMMENTS

Referee #1:

The authors have properly addressed all my comments.

Referee #2:

The authors have satisfactorily addressed all of my previous comments. The revisions are clear, appropriate, and effectively resolve the concerns raised in my earlier review.